

# An update of shallow cloud parameterization in the AROME NWP model

Adrien Marcel[1], Sébastien Riette[1], Didier Ricard[1], and Christine Lac[1]

[1]CNRM, Université de Toulouse, Météo-France, CNRS, Toulouse, France

**Correspondence:** Adrien Marcel (adrien.marcel@meteo.fr), Sébastien Riette (sebastien.riette@meteo.fr), Didier Ricard (didier.ricard@meteo.fr), and Christine Lac (christine.lac@meteo.fr)

**Abstract.** The representation of shallow clouds in numerical weather prediction models remains a challenge for the parameterizations of Atmospheric Boundary Layer (ABL). Previous evaluations of the AROME model have shown radiative budget weaknesses, which were later attributed to a lack of shallow clouds, especially stratocumulus and small cumulus. In this study, we investigate the difficulties of the AROME model to represent the ABL and the associated low clouds, and we provide con-
sistent updates of the Eddy Diffusivity Mass Flux (EDMF) scheme (shallow convection plus turbulence schemes), the subgrid cloud scheme and the associated precipitation. For this purpose, we use the well-known Single Column Model (SCM) versus Large Eddy Simulation (LES) comparison approach to evaluate the modifications. Additionally, the semi-automatic HighTune Explorer tool helps us to explore the free parameter space associated with the modified parameterizations. The modifications are evaluated using several documented cases of boundary layer development, from stratocumulus to precipitating cumulus clouds, including a transition case. Although physical inconsistencies and questionable assumptions still remain in AROME
and need to be clarified, the modifications improve the consistency between parameterizations and yield simulations in better agreement with LES, in particular for the diurnal cycle of clouds and the representation of non-local turbulence.

## 1  Introduction

The importance of subgrid process parameterizations in the Atmospheric Boundary Layer (ABL) has been recognized for
decades. They are now ubiquitous in atmospheric modelling for both forecasting and climate applications (Madeleine et al., 2020; Doms et al., 2021; ECMWF, 2024), yet they are still one of the main sources of error identified in large-scale sensitivity assessments (Betts and Jakob, 2002; Dufresne and Bony, 2008; Shin and Dudhia, 2016; Mantovani Júnior et al., 2023). Increasing computing power means that Numerical Weather Prediction (NWP) models can now represent finer turbulent structures for operational applications. Today's NWP models can resolve down to one kilometre for local forecasts (e.g. Brousseau et al.,
2016), and around ten kilometres for global systems (e.g. Malardel et al., 2016). At kilometric resolution, deep convection is considered resolved, but shallow convection in the ABL and just above still needs to be parameterized. Therefore, parameterization improvements remain important, as subgrid processes are still among the main sources and sinks of conservative quantities in the ABL.





Historically, several approaches have been used for this purpose. one popular and simple design is the K-gradient closure, which aims to describe small-scale turbulence in the form of isotropic eddies. This approach works relatively well at resolutions outside the ABL turbulence spectrum, e.g. at coarse mesoscale resolution or at the LES scale. However, at kilometric and mostly sub-kilometric horizontal resolution ("Terra Incognita" for shallow convection Wyngaard (2004)), it does not work very well to reproduce the ABL counter-gradient regions. Later, Deardorff (1966) introduced a modified K-gradient formulation
using a counter-gradient constant, which is very easy to use in practice and has shown relatively good but inherently limited performance. Such a simple closure presents physical and conceptual deficiencies (Zhou et al., 2018), especially in the upper part of the ABL. For atmospheric modeling, more sophisticated Higher-Order Closures (HOC) are popular and employed as well (Canuto et al., 1994; Tomas and Masson, 2006). These formulations require additional prognostic equations that are computationally intensive, or have been developed for a specific purpose (e.g., only for dry ABL).

The 'Eddy Diffusivity Mass Flux' framework has enjoyed considerable success over the past few decades. It has been used in both atmospheric (Arakawa and Schubert, 1974; Hourdin et al., 2002; Cheinet, 2003; Soares et al., 2004; Siebesma et al., 2007; Neggers et al., 2009; Neggers, 2009; Sušelj et al., 2013; Olson et al., 2019) and oceanic contexts (Giordani et al., 2020; Garanaik et al., 2023; Perrot et al., 2025; Perrot Manolis, 2024) (not exhaustive). It aims to represent ascending or sinking
thermal plumes regions with a surrounding turbulent environment. Several versions of EDMF have been proposed. The most commonly used are the bulk {updrafts-environment} and bulk {updrafts-downdrafts-environment} frameworks. Implementations of multiple EDMF subdomains have also been developed and integrated in Cheinet (2003); Tan et al. (2018) or Suselj et al. (2019) for example. Efforts have also been made to develop unified parameterizations for ABL processes. Many of these approaches use the EDMF concept to represent the isotropic and anisotropic parts of the turbulence. These have been developed
for one-, two- or higher-order closure schemes (Lappen and Randall, 2001a, b, c).

EDMF has been shown to improve long standing issues in a consistent way to unify all ABL regimes (Kurowski et al., 2019). However, closure uncertainties remain associated with the concept. Critical parameterizations concern entrainement and detrainment, for example. They have been the subject of a great deal of research over the last decades. The relevance of surround-
ing air entraining in convective updraughts has been established by Stommel (1947). Since then, many studies on entrainment and detrainment have been carried out using either simulations (Chosson et al., 2007) or observations (Burnet and Brenguier, 2007). Phase change also plays an important role in triggering evaporative cooling of subsiding parcels near the cloud edges (Paluch, 1979). Nevertheless, it remains very challenging to quantify these processes. An overview of the main modeling ideas of entrainment and detrainment are exposed in de Rooy et al. (2012). Difficulties on closures also hinder parameterization de-
velopment, for example with regard to pressure perturbation terms in the momentum budgets. However, the parameterizations literature (and EDMF) is quite extensive, and efforts have been made to include radiation, microphysics processes (Suselj et al., 2022), energy conservation (Perrot et al., 2025) and a better consideration of cloud-related subgrid effects (Cohen et al., 2020).



Parameterization development is computationally intensive. Therefore, the use of Single Column Model (SCM) is ubiquitous
in the community. However, it does not provide a complete representation of a NWP model. The evaluation of SCM models is
typically carried out by comparing them to LES references in well-known documented cases of ABL, or with direct observa-
tions. In parallel, the calibration of free parameters is also difficult when there are many of them. To improve model physics,
Couvreux et al. (2021); Hourdin et al. (2021) proposed machine learning techniques with a calibration tool: HighTune explorer
(HTexplo), to carefully tune multiple free parameters. The HTexplo tool has already been tested in climate model studies, such
as the LMDz6A IPSL model or the Meteo-France global NWP ARPEGE model for very stable boundary layers (Audouin
et al., 2021).

In this study, our objective is to improve the area-limited AROME model (Seity et al., 2011) in its 1.3 km horizontal resolution
configuration. Recent evaluations of the AROME NWP model have highlighted deficiencies in its representation of the ABL.
Systematic biases in certain cloud types have been detected in annual measurements of surface incident solar radiation over
France. Some of the largest positive biases for low clouds were later attributed to specific types of stratified clouds, such as
stratocumulus, through the use of large aperture sky cameras and in situ measurements, such as the Meteopole Flux (Cal-
vet et al., 2016; Canut et al., 2019) platform at the CNRM laboratory site in Toulouse. Additionally, a more comprehensive
radiative analysis performed by Magnaldo et al. (2024) shows the biases as a function of a cloud classification in the model.
Furthermore, studies point to variability in the model's response to convective situations (e.g. Riette and Lac, 2016), suggesting
that further investigations are needed to identify the critical physical processes in the ABL that mainly cause this variability.
Stratocumulus clouds are challenging to model in NWP and climate models, yet their representation is crucial given their
ubiquity in the ABL. For example, Köhler et al. (2011) established the key ingredients for a successful stratocumulus scheme.
Wood (2012) provided a comprehensive overview of the most important processes involved in the Stratocumulus Boundary
Layers (ScBL). They are a subtle balance between turbulence, cloud microphysics, and radiation; in particular, the cloud top
entrainment and radiative cooling effects are critical for ScBL development.

The main objective of this paper is thus to investigate and update the AROME ABL parameterizations, including modifications
to the shallow convection, turbulence, cloud and microphyics schemes in a consistent manner. The modifications will be
evaluated using an SCM versus LES framework on ABL cases. Free parameters will be tuned using the HTexplo tool. The paper
is organized as follows: LES-1D methodology (section 2.1), a brief description of the AROME parameterizations (section 2.2),
a description of the High-Tune explorer tool (section 2.3), and an update of the AROME parameterization schemes for the
ABL focusing on cumulus and stratocumulus clouds (section 3). The results are presented in section 4, after implementation
and free parameters tuning of the modifications. The paper concludes with a discussion of future work on the model and open
issues.



## 2 Methodology

### 2.1 LES versus SCM setup

This study examines four well-documented cases of low-level clouds in the ABL. The ARMCu cumulus cloud is an idealized shallow convective cloud, formed during the Atmospheric Radiation Measurement (ARM) campaign on 21 June 1997 in the
Great Plains region of the United States (Brown et al., 2002). The RICO case is an idealized shallow precipitating cumulus cloud over the Atlantic Ocean, based on observations collected during the Rain in Cumulus over the Ocean field study (van-Zanten et al., 2011) near the islands of Antigua and Barbuda. The FIRE case is a marine stratocumulus topped boundary layer, defined by Duynkerke et al. (2004) based on observations from the First ISCCP Regional Experiment and the EUROpean Cloud Systems (EUROCS) project. The SANDU case is an idealized stratocumulus-to-cumulus cloud transition described in Sandu
and Stevens (2011), which is a stratocumulus-topped deepening boundary layer, driven mainly by sea surface temperature (SST) in the north-eastern Pacific.

From these idealized cases, we have carried out a number of LES using the Meso-NH research model (Lac et al., 2018). As the physics of AROME comes from Meso-NH, it should be noted that the underlying physics of Meso-NH LES is really close to that of AROME with only minor differences in the schemes: basically, the activated radiation and microphysical scheme
are the same between AROME and the performed LES. In Meso-NH LES, the shallow convection scheme is disabled and the turbulence scheme is a modified version of the AROME scheme, including horizontal fluxes. Meso-NH offers the opportunity to run a wide range of offline and online diagnostics (e.g., coarse graining and energy budgets) and the conditional sampling method from Couvreux et al. (2009) is implemented. This method uses passive radioactive tracers to recover simulated objects in the LES grid. In parallel, all cases are run with an SCM version of AROME (Malardel, 2008), which serves as a basis for
developing new parameterizations and implementing modifications with reasonable time and resources. The AROME SCM requires input profiles and large-scale forcings to be supplied in a standard format defined by the DEPHY community (see: https://github.com/GdR-DEPHY/DEPHY-SCM).

### 2.2 AROME / Meso-NH parameterization framework

AROME is a flexible, limited-area NWP model that can be adapted to different domains and resolutions. In operational use at Météo-France, AROME achieves a horizontal resolution of 1.3 km and has 90 vertical levels from the surface to the strato-sphere layer at approximately 27.5 km. Meso-NH is a non-hydrostatic mesoscale research model that deals with scales ranging from large (synoptic) to small (large eddy). AROME and Meso-NH have different dynamical cores, but they use similar physics (Seity et al., 2011). More details on general aspects of the physical framework can be found on the Meso-NH website
(http://mesonh.aero.obs-mip.fr/mesonh57/BooksAndGuides). A radiation scheme based on the ECMWF for shortwave radiation from Fouquart and Bonnel (1980) and the Rapid Radiative Transfer Model (RRTM) for longwave radiation from Mlawer et al. (1997) is applied. Additionally, the surface fluxes are computed with an external platform called SURFEX (Masson et al., 2013), a surface software developed at Météo-France, CNRM. The next subsections provide a brief description of the other





implemented physics.


### 2.2.1 Eddy-Diffusivity Mass-Flux (EDMF) approach

Focusing on the representation of AROME turbulence and shallow convection, the conservative quantities used are the enthalpy (using the pseudo-conservative liquid-ice water potential temperature $\theta_{il} = \theta - \frac{L_v}{C_{pm}\Pi}r_c - \frac{L_s}{C_{pm}\Pi}r_i$, $\Pi$ is the Exner function, $L_v$ and $L_s$ are the latent heat of vaporization and sublimation respectively, $C_{pm}$ is the specific heat of moist air at constant

pressure and $r_c$, $r_i$ are the mixing ratios of liquid cloud water and ice respectively), the mass (using the total water mixing ratio $r_t = \sum_j r_j$, $r_j$ are the mixing ratios of the non-precipitating water species) and the momentum $\overrightarrow{v} = (u,v,w)$. Their tendencies require the application of subgrid parameterized turbulent fluxes using Reynolds averaged Navier-Stockes equations. In both the Meso-NH and AROME models, a framework combining Eddy Diffusivity (ED) and Mass Flux (MF) is used to account for both local and non-local mixing respectively in the ABL. It assumes an isotropic turbulence environment for

the 'ED' part, and a bulk single updraft assumption inherited from Arakawa and Schubert (1974) for the 'MF' part. The EDMF approach involves dividing the horizontal grid into several distinct spatially and statistically averaged objects. Given the aforementioned considerations and the small area limit ($a_u << 1$ where $a_u$ is the updraft fraction), all vertical turbulent fluxes can be approximated as follows:

$$\overline{w'\phi'} \simeq \underbrace{-K\frac{\partial\overline{\phi}}{\partial z}}_{\text{ED}} + \underbrace{a_u\overline{w}_u(\overline{\phi}_u - \overline{\phi})}_{\text{MF}}, \quad \phi \in \{\theta_{il}, r_t, u, v\} \tag{1}$$

The Reynolds decomposition implies that, for a variable $\phi = \overline{\phi} + \phi'$, where $\overline{\phi}$ is the grid average of $\phi$, and $\phi'$ is a fluctuation around the mean such that $\overline{\phi'} = 0$. The '**u**' subscript refers to the 'updraft' variables, K is a diffusivity constant, $w$ is the vertical velocity. In AROME, the MF terms are computed in the shallow convection scheme and the ED terms are computed in the turbulence scheme. The aim is to obtain an ensemble picture of grid-averaged fluxes derived from an EDMF framework.

### 2.2.2 Turbulence scheme (CBR)

Both AROME and Meso-NH use an 1.5 prognostic Turbulent Kinetic Energy (TKE) equation closure scheme from Cuxart et al. (2000) (CBR). Using the Einstein summation notation ($i,j \in \{1,2,3\}$), the prognostic TKE ($e = \frac{1}{2}\overline{u_i'^2}$) equation reads as follows:

$$\frac{\partial e}{\partial t} = \underbrace{-\frac{1}{\rho_{\text{ref}}}\frac{\partial}{\partial x_j}(\rho_{\text{ref}}e\overline{u_j})}_{\text{Advection}} \underbrace{-\overline{u_i'u_j'}\frac{\partial\overline{u_i}}{\partial x_j}}_{\text{Shear}} + \underbrace{\frac{g}{\theta_{v_{\text{ref}}}}\overline{u_3'\theta_v'}}_{\text{Buoyancy}} + \underbrace{\frac{1}{\rho_{\text{ref}}}\frac{\partial}{\partial x_j}\left(\rho_{\text{ref}}\overline{u_j'\frac{u_i'^2}{2}} + \overline{u_j'p'}\right)}_{\text{Turbulent transport}} - \underbrace{\epsilon_e}_{\text{Dissipation}} \tag{2}$$

In the CBR scheme, this relation is closed by setting $\overline{u_j'\frac{u_i'^2}{2}} = C_{2m}L_m e^{\frac{1}{2}}\frac{\partial e}{\partial x_j}$, $\epsilon_e = C_{\text{diss}}\frac{e^{\frac{3}{2}}}{L_\epsilon}$, and $\overline{w'p'} = 0$ where $C_{2m}$ and

$C_{\text{diss}}$ are constants for turbulent transport and dissipation, g is the acceleration due to gravity, $L_m$ and $L_\epsilon$ are the mixing and dissipation length scales respectively, and $\rho_{\text{ref}}$ and $\theta_{v_{\text{ref}}}$ are the reference density and reference virtual potential temperature,





respectively. In AROME, $L_m = L_\epsilon$ is the Bougeault-Lacarrere length BL89 (Bougeault and André, 1986; Bougeault and Lacarrere, 1989). Only the vertical components of the equation 2 (except for the advection of TKE for which the three components are taken into account) are implemented in AROME, whereas all components are considered in Meso-NH. This corresponds to a 1D turbulence scheme. In this scheme, the diffusivity constants for the quantity $\phi$ are expressed as $K_\phi = C_\phi L_m e^{\frac{1}{2}}$, where $C_\phi$ is a constant for a specified diagnosed flux.

### 2.2.3 Shallow convection scheme (PMMC09)

The shallow convection scheme of Pergaud et al. (2009) PMMC09 calculates a diagnostic version of the MF contribution. The mass flux ($M_u$), updraft vertical velocity ($\overline{w}_u$), updraft fraction ($a_u$) and updraft properties are essential for accounting for the dry and wet components of the ABL and read with the steady-state assumption:

$$
\begin{cases}
\frac{1}{M_u}\frac{\partial M_u}{\partial z} = \epsilon - \delta \\[2mm]
\overline{w}_u \frac{\partial \overline{w}_u}{\partial z} = aB_u - b\epsilon\overline{w}_u^2 \\[2mm]
a_u = \frac{M_u}{\rho_{\text{ref}}\overline{w}_u} \\[2mm]
\partial_z \overline{\phi}_u = -\epsilon(\overline{\phi}_u - \overline{\phi}) + S_{u,\phi}, \quad S_{u,\phi} = \begin{cases} 0 & \text{if } \phi \in \{r_t, \theta_{il}\} \\ C_\phi \partial_z \overline{\phi} & \text{if } \phi \in \{u,v\} \end{cases}
\end{cases}
\tag{3}
$$

where $\epsilon$ and $\delta$ are the fractional entrainment and detrainment, respectively ($E = M_u\epsilon$, $D = M_u\delta$, with $E$ and $D$ being the entrainment and detrainment), $B_u$ is the buoyancy of the ascending updraft, and {a,b} are closure parameters. $C_u$ and $C_v$ are coefficients extracted from Gregory et al. (1997) and Kershaw and Gregory (1997) to represent horizontal pressure gradient adjustments for the updraft horizontal velocities $u_h = (u,v)$.

In the bulk single updraft EDMF approach, parameterization requires fractional entrainment and fractional detrainment quantity closures to control the mass flux through Eq. 3. Numerous studies have demonstrated links between $\epsilon$, $\delta$ and $\frac{B_u}{w_u^2}$ (Nordeng, 1994; Gregory, 2001; de Rooy and Siebesma, 2010; Rio et al., 2010). The current scheme uses the following formulations for $\epsilon$ and $\delta$ in the dry ABL:

$$
\epsilon_{\text{dry}} = \text{Max}\left(0, C_{\epsilon_{\text{dry}}}\frac{B_u}{w_u^2}\right)
$$

$$
\delta_{\text{dry}} = \text{Max}\left(\frac{1}{L_{\text{up}}(z_{\text{srf}}) - z}, C_\delta \frac{B_u}{w_u^2}\right), \quad \int_{z_{\text{srf}}}^{z_{\text{srf}}+L_{\text{up}}(z_{\text{srf}})} \frac{g}{\theta_{v_{\text{ref}}}}\left(\theta_v(z') - \theta_v(z)\right)dz' = -e(z_{\text{srf}})
\tag{4}
$$

$C_{\epsilon_{\text{dry}}}$ and $C_\delta$ are closure constants, and $\theta_v$ is the virtual potential temperature. It is assumed that the updraft entrains environmental air near the surface layer, where $B_u$ is strong and detrain when it reaches convective inhibition CIN regions, or the top of the ABL in dry conditions. Several studies also suggest that $\delta$ is not zero in the dry part of the ABL. Then, Pergaud et al. (2009) added a minimum fractional detrainment $\frac{1}{L_{\text{up}}(z_{\text{srf}}) - z}$ to account for this effect. This was extracted and adapted from the ADHOC parameterization by Lappen and Randall (2001b), in which the quantities $\epsilon$ and $\delta$ are parameterized as being inversely proportional to a characteristic length. This analogy originates from the mean-variance prognostic equations, in which





the quantity $\epsilon + \delta$ acts as a dissipation term (see de Roode et al. (2000); Lappen and Randall (2001a)). Thus, entrainment and detrainment are known to dissipate convective plumes. A dissipation rate can be easily represented by a suitable characteristic length ($L_c$) or timescale ($\tau_c$), such as $\partial_t \overline{\phi'\psi'} \sim \frac{v_c \overline{\phi'\psi'}}{L_c}$ ($v_c$ is a characteristic velocity). In the formulation, $L_{\mathrm{up}}(z_{\mathrm{srf}})$ is the up-
ward component of the BL89 length, which is also used in CBR and computed at the surface.

The wet part representation is more complicated. To account for the condensed water effects, PMMC09 uses a modified buoyancy sorting scheme from Kain and Fritsch (1990) assuming a uniform PDF for the mass distribution from Bretherton et al. (2004). For entrainment and detrainment, it reads:

$$E = \Delta M_t \chi_c^2$$
$$D = \Delta M_t (1 - \chi_c)^2 \tag{5}$$
$$\Delta M_t = \frac{C_{w_{\epsilon+\delta}} \Delta z}{R}$$

$\Delta M_t$ is the mixed mass at the boundary between the updraft and the environment domain, $\chi_c$ is the critical environmental fraction giving a neutrally buoyant mixture, R is the updraft radius, and $C_{w_{\epsilon+\delta}}$ is a closure constant. $\chi_c$ is solved directly using a linear approximation, knowing that the virtual potential temperature $\theta_v$ difference between the plume - environment mixture, $\theta_{v_{\mathrm{mix}}}$, and that of the unmixed environment, $\overline{\theta}_v$, varies linearly with $\chi$. The zero crossing of the function is evaluated from:

$$\chi_c = \frac{\theta_{v_u} - \overline{\theta}_v}{\theta_{v_u} - \theta_{v_{\mathrm{mix}}}} \chi, \quad \chi = 0.1 \tag{6}$$

assuming that $\chi < \chi_c$. Finally, Pergaud et al. (2009) added a control to prevent the wet entrainment to be larger than the wet detrainment such as:

$$E = \Delta M_t \mathrm{Min}[\chi_c^2, (1 - \chi_c)^2] \tag{7}$$

PMMC09 is discretized and computed vertically from the bottom to the top level of the model. It uses the following closure formulation at the ground level:

$$\begin{cases} M_u(z_{\mathrm{srf}}) = \mathrm{C}_{\mathrm{M}_0} \rho_{\mathrm{ref}} \left( \frac{g}{\theta_{v_{\mathrm{ref}}}} \overline{w'\theta'}_{v\,\mathrm{srf}} L_{\mathrm{up}}(z_{\mathrm{srf}}) \right)^{\frac{1}{3}} \\ \overline{\phi}_u = \overline{\phi}(z_{\mathrm{srf}}) + \alpha_{\phi_{\mathrm{srf}}} \frac{\overline{w'\phi'}_{\mathrm{srf}}}{\sqrt{e(z_{\mathrm{srf}})}} \\ \overline{w}_u^2(z_{\mathrm{srf}}) = \frac{2}{3} e(z_{\mathrm{srf}}) \end{cases} \tag{8}$$

where $\mathrm{C}_{\mathrm{M}_0}$ and $\alpha_{\phi_{\mathrm{srf}}}$ are model parameters.

### 2.2.4   Cloud scheme

The subgrid saturation adjustment cloud condensation scheme is based on the saturation deficit variable $s$, firstly proposed by Mellor (1977) for warm phase, and extended to mixed-phase clouds for Meso-NH by Chaboureau and Bechtold (2002)
(CB02):

$$\begin{cases} s = c(r_t - r_{\mathrm{sat}_{il}}(T_{il})) \\ c = \frac{1}{1 + \left( \frac{\partial r_{\mathrm{sat}_{il}}}{\partial T} |_{T_{il}} \right) \frac{L}{C_{pm}}}, \quad \frac{\partial r_{\mathrm{sat}_{il}}}{\partial T} |_{T_{il}} \simeq \frac{r_{\mathrm{sat}_{il}} L}{R_v T_{il}^2} \left( 1 + \frac{r_{\mathrm{sat}_{il}} R_v}{R_d} \right) \end{cases} \tag{9}$$





where $T_{il} = \Pi\theta_{il}$, $r_{\mathrm{sat}_{il}}$ is the averaged saturation mixing ratio between the saturation mixing ratios of liquid and ice water, with the same treatment for $L$. $R_v$ and $R_d$ are the gas constants for water vapor and dry air, respectively. $Q_1$ is then defined as the value of s normalised by its variance. The turbulence scheme provides statistical information on the subgrid distribution of

$r_t$ and $\theta_{il}$. Thus, the environmental saturation deficit variance, $\overline{s_{\mathrm{ED}}'^2}$, is diagnosed from the pseudo-conservative thermodynamic variances $(\overline{r_t'^2}, \overline{\theta_{il}'^2})$ and co-variance $(\overline{r_t'\theta_{il}'})$, as $\overline{s_{\mathrm{ED}}'^2} = \bar{c}^2\overline{r_t'^2} + 2\overline{d}\bar{c}\overline{r_t'T_{il}'} + \overline{d}^2\overline{T_{il}'}$, with $d = c\frac{\partial r_{\mathrm{sat}}}{\partial T}|_{T_{il}}$. In the CB02 scheme, $Q_1$ is used to diagnose the cloud quantities $CF$, $\overline{r_c}$ and $\overline{s'r_c'}$ with analytical formulations. The model also assumes that the adjustment to the saturation process is instantaneous at each time step.

Additionally, the updraft-driven cloud of shallow convection scheme is diagnosed 'directly', assuming that all updraft water

vapor $\overline{r_{vu}}$ exceeding the updraft saturation $\overline{r_{\mathrm{sat}u}}$ is converted to updraft cloud condensate, $\overline{r_{cu}}$. The shallow cloud fraction contribution $(CF_u)$ is then estimated proportionally from the updraft fraction, $CF_u = C_{cf}a_u$ ($C_{cf}$ being a model parameter), and the shallow cloud condensate contribution is then given by $\overline{r_c} = CF_u\overline{r_{cu}}$.

The total subgrid cloud contribution is the sum of these two contributions. The interactions between water species are then treated in the microphysical scheme.

### 2.2.5    Microphysical scheme (ICE3)

AROME uses a one-moment microphysical scheme (ICE3) for ice-water species interactions from Pinty and Jabouille (1998) coupled to the warm cloud microphysical processes of Kessler (1995). It uses prognostic equations for the mixing ratios of water vapour $r_v$, cloud liquid water $r_c$, rain $r_r$, cloud ice water $r_i$, snow $r_s$ and graupel $r_g$. Only a brief description is provided here for key processes related to ABL clouds. In ICE3, the autoconversion processes (conversion of cloud species into rain and

snow) are parameterized based on the consideration that they increase linearly with the water content above a threshold:

$$\begin{cases} P_{\mathrm{aut}_{RC}} = k\mathrm{Max}\left(0, r_c - \frac{\rho_{c_{\mathrm{crit}}}}{\rho_{\mathrm{ref}}}\right) \\ P_{\mathrm{aut}_{RI}} = k_{is}\mathrm{Max}(0, r_i - r_{i_{\mathrm{crit}}}), \ r_{i_{\mathrm{crit}}} = \mathrm{Min}\left(2.10^{-5}, 10^{0.06.(T-T_t)-3.5}\right) \end{cases} \tag{10}$$

$P_{\mathrm{aut}}$ refers to the tendency of the process, $k_{is} = 10^{-3}e^{0.015(T-T_t)}$ and $k$ are the inverse proportional time constants for pristine ice and cloud droplets, respectively ($T$ is the temperature and $T_t$ is the triple point temperature of water), and the subscript **'crit'** refers to the parameterized thresholds on both cloud liquid water and pristine ice species. Additionally, liquid water

autoconversion uses a uniform PDF to represent variability in cloud content within the model cell (Redelsperger and Sommeria, 1986). Its width is equal to the variance of the cloud scheme's saturation deficit variable; the cloud fraction is not used. Evaporation of rain droplets is considered using Pruppacher and Klett (1978) formulations.

### 2.3    HighTune Explorer Tool

The set of parameterizations includes a significant number of closure parameters. Most of these can be explained by a physical

interpretation, providing insight into their values. However, it is uncertain whether model bias and the failure to represent the underlying physical phenomena ensure total agreement between the expected physical value and the effective value that best represents the subgrid fluxes. Additionally, a manual calibration of the free parameters is difficult due to the many degrees





of freedom. Even for a few parameters, let's say 5, we would need $10^5$ simulations with an SCM framework, if we sampled the range of possible values for each free parameter with 10 values. Here, we use surrogate models to sample and explore the associated N-parameter space appropriately, using the HighTune explorer tool (HTexplo), which has already proven particularly useful in the parameterization community. A detailed description of the tool can be found in Couvreux et al. (2021) and Hourdin et al. (2021). It works by exploring and resampling the parameter space NROY$^0$ (The initial **N**ot **R**uled **O**ut **Y**et space) associated with a selected number of free parameters, using history matching (Williamson et al., 2013) with iteratively refocusing waves. Each wave discards, from the initial parameter space, the parameter combinations that produce simulations too far from the reference LES (using a set of metrics). A brief description of the HTexplo steps can be found in Appendix A.

## 3 AROME updates

We decided to update the AROME subgrid physics while maintaining the bulk {updraft - environment} approach for the shallow convection scheme.

### 3.1 Preliminary changes

Before adding physical elements to the code, preliminary investigations were carried out to identify a number of problems, most of which are purely algorithmic. Additionally, the choice made by Pergaud et al. (2009) to prevent the entrainment rate in the cloud from exceeding the detrainment rate caused a major problem (see Eq. 7).

Figure 1 shows that removing this constraint has no effect on clouds such as ARMCu. However, it greatly modifies the behaviour of wet mass flux in certain cases, such as with FIRE stratocumulus clouds, for which the diurnal cycle is better reproduced. The simulation incorporating these preliminary changes is referenced as the CTRL experiment and will serve as a control reference for the rest of this study.





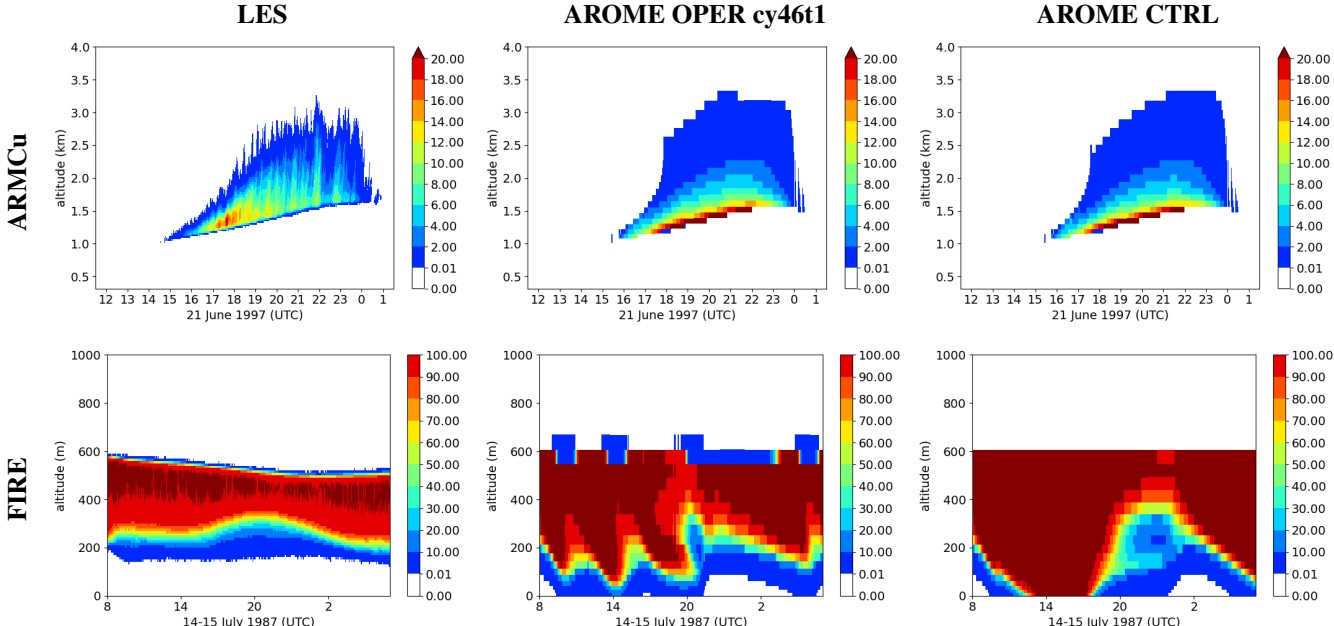

**Figure 1.** Temporal evolution of the vertical profiles of cloud fraction (CF) [%] for the ARMCu case (first line), and the FIRE case (second line) with the LES (first column), the operational AROME version (second column), and the CTRL reference experiment (third column).

## 3.2 Shallow convection scheme

### 3.2.1 Small updraft fraction assumption removal

The AROME shallow convection scheme uses horizontal grid averaging according to the EDMF framework. In this context,
255   the horizontal mean $\overline{\phi}$ of a variable $\phi$ can be written as $\overline{\phi} = (1 - a_u)\overline{\phi}_e + a_u\overline{\phi}_u$, $\overline{\phi}_e$ is the horizontally averaged properties over
the environmental subdomain. Therefore, the general form of grid-mean variances and covariances are for any variable $\phi$ and
$\psi$:

$$\overline{\phi'\psi'} = (1 - a_u)\overline{\phi'_e\psi'_e} + a_u\overline{\phi'_u\psi'_u} + (1 - a_u)(\overline{\phi}_e - \overline{\phi})(\overline{\psi}_e - \overline{\psi}) + a_u(\overline{\phi}_u - \overline{\phi})(\overline{\psi}_u - \overline{\psi})$$

A particular case where $\psi = w$ and simplifying leads to: $\hspace{3cm}$ (11)

260   $$\overline{w'\phi'} = (1 - a_u)\overline{w'_e\phi'_e} + a_u\overline{w'_u\phi'_u} + \underbrace{(1 - a_u)\overline{w}_e(\overline{\phi}_e - \overline{\phi})}_{\frac{M_e}{\rho}} + \underbrace{a_u\overline{w}_u(\overline{\phi}_u - \overline{\phi})}_{\frac{M_u}{\rho}}$$

assuming that $\overline{w} = 0$.

The classical version of EDMF often assumes a small updraft area compared to the horizontal resolution of the model grid.
For shallow convection, and given the operational resolution of AROME, there is no evidence that the classical assumption
$a_u << 1$ used in Eq. (1) is still valid (Honnert et al., 2016). Thus, we decided to remove this assumption for the mass flux part.
265   Without additional subdomain assumptions and rewriting the environmental mass flux contribution from Eq. (11), the total





mass flux variance-covariance contribution (the last two terms of Eq. (11)) becomes:

$$\overline{\phi'\psi'}_{MF} = \frac{a_u}{1-a_u}(\overline{\phi}_u - \overline{\phi})(\overline{\psi}_u - \overline{\psi}) \tag{12}$$

The term $\overline{\phi'_e\psi'_e}$ is calculated directly by the CBR scheme. Note that $(1-a_u) \simeq 1$ for turbulence is not modified, so the assumption $a_u\overline{\phi'_u\psi'_u} + (1-a_u)\overline{\phi'_e\psi'_e} \simeq \overline{\phi'_e\psi'_e}$ is considered, although a simple formulation for $\overline{\phi'_u\psi'_u}$ could be added. In Equation (3), the updraft scalar uses the approximation $\overline{\phi}_e \simeq \overline{\phi}$. Relaxing this assumption, it becomes:

$$\partial_z\overline{\phi}_u = -\frac{\epsilon}{1-a_u}(\overline{\phi}_u - \overline{\phi}) + S_{u,\phi}, \quad S_{u,\phi} = \begin{cases} 0 & \text{if } \phi \in \{r_t, \theta_{il}\} \\ C_\phi\partial_z\overline{\phi} & \text{if } \phi \in \{u, v\} \end{cases} \tag{13}$$

### 3.2.2 Updraft vertical velocity parameterization

One of the main difficulties encountered with EDMF systems is the parameterization of the mean updraft vertical velocity. Given the prognostic equations for the updraft properties (see Tan et al. (2018) for example), it is possible to rewrite the mean updraft vertical velocity equation as well. With the assumption $\overline{w} = 0$ suggested by the continuity equation in SCM mode, we have:

$$\partial_t\overline{w}_u + \underbrace{\frac{1}{2}\partial_z\overline{w}_u^2 + \frac{\partial_z\overline{a_u w'^2_u}}{a_u}}_{\text{Advection}} = B_u - \underbrace{\partial_z\left(\frac{\overline{p}^\dagger}{\rho_{\text{ref}}}\right)_u}_{\text{Forces}} - \underbrace{\left(\frac{\epsilon}{1-a_u}\overline{w}_u^2\right)}_{\text{Entrainment}} \tag{14}$$

Where $p^\dagger = p - p_{\text{ref}}$ and $p_{\text{ref}}$ is the reference hydrostatic pressure. Following PMMC09, the stationary equilibrium assumption is still used for the AROME-EDMF equations. Perrot et al. (2025) argue that this remains valid as long as the surface forcing evolves slowly compared to the atmospheric stratification, which is verified in most cases for shallow convection. Recently, the parameterization community has shown an interest in the pressure term $\partial_z\left(\frac{\overline{p}^\dagger}{\rho_{\text{ref}}}\right)_u$ closure. Its dynamical and thermodynamical contribution appears to be non-negligible for both shallow and dry thermals, as suggested by Gu et al. (2020) and Morrison et al. (2022). Some studies have attempted to make it prognostic by making strong assumptions about the updraft form factor in an idealized SCM model (e.g. Leger et al., 2019). Other studies have proposed diagnostics to evaluate this term, such as Morrison and Peters (2018); Peters et al. (2021). In this work, we have opted to adopt the formulation of He et al. (2020), which is built on single normal mode solutions based on 2D thermals and 3D axisymmetric thermals. Assuming the pressure term is proportional to buoyancy, with an additional advective and drag contribution, and considering the turbulent transport term $\frac{\partial_z\overline{a_u w'^2_u}}{a_u}$ to be negligible, we thus have:

$$(1-\alpha_a)\partial_z\overline{w}_u^2 = 2aB_u - \frac{2b\epsilon}{1-a_u}\overline{w}_u^2 - \frac{2b'}{H(1-a_u)^2}\overline{w}_u^2 \tag{15}$$

(using the relation $w_u - w_e = \frac{w_u - \overline{w}}{1-a_u}$) where $aB_u$ is an "effective buoyancy", $b$ is a correlation coefficient between pressure perturbation and fractional entrainment, $b'$ is a drag term coefficient, $\alpha_a$ is an advective coefficient, and $H$ is a characteristic drag length.



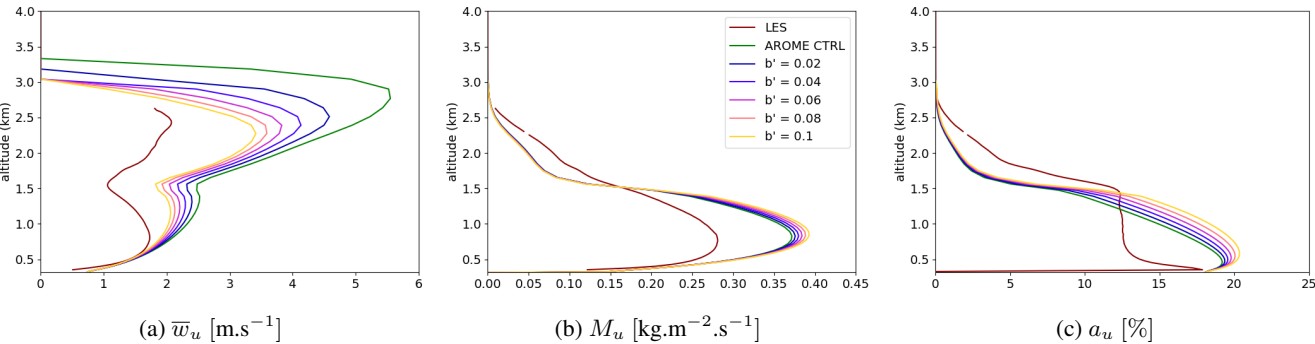

**Figure 2.** Effect of the drag parameterization on the vertical velocity (last term in Eq: 15) of AROME from ARMCu case: (a) mean updraft vertical velocity, (b) updraft mass flux and (c) updraft fraction with variation in $b'$ parameter (colors), LES conditional sampling (red line) and the CTRL experiment ($b' = 0$, green line).

The introduced quantity $H$ requires a closure because all terms of Eq. (15) are inversely proportional to a length. Simple formulations also exist, such as that in Rio et al. (2010) which has already produced good results in GCMs, for example. Their drag coefficient, $b'$, is a constant that is inversely proportional to a length scale and already includes $H$. Therefore, it is not dependent on the ABL regime, although an updraft height or radius dependency would be natural. We use the Tan et al. (2018) formulation, where $H$ is diagnosed directly using an approximated updraft radius, assumed to be spherical, such as $H = r_d\sqrt{a_u}$, where $r_d$ is the characteristic horizontal spacing between the plumes. Figure 2 shows the effect of the drag parameterization on the ARMCU case. The formulation can estimate a pressure drag effect for a cumulus cloud, which is strongest at the cloud top and near the lifting condensation level (LCL). In accordance with He et al. (2020), the drag seems to be underestimated at the bottom of the ABL. The impact of drag parameterization on shallow convection is significant, as $a_u$ varies by a few percents between $b' = 0.1$ and the drag-free case.

### 3.2.3 Modified entrainment and detrainment

Another update concerns the fractional entrainment, $\epsilon$, and detrainment, $\delta$. In dry conditions, a good estimation of fractionnal detrainment and entrainment is $\frac{B_u}{w_u^2}$ (see Eq. 4), which is consistent with the analytical solution in de Rooy and Siebesma (2010) and other parameterization frameworks based on Gregory (2001) or Nordeng (1994).

For the dry part of the ABL in PMMC09, a minimum fractional detrainment is provided in the form of $\frac{1}{L_{up}(z_{srf})-z}$, which can be used to estimate the updraft height above altitude z. Note that the $L_{up}$ formulation does not take entrainment, detrainment, or phase changes into account. In the plume, we can partially correct the effects of water phase changes by using the plume properties at altitude z. However, it is difficult to fully correct this approximation in the current scheme, as it is built from the surface to the top of the ABL. Its properties above level z are unknown. Therefore, we have decided to evaluate the minimum fractional detrainment with the $L_{up}(z)$ length using the updraft properties at level $z$ instead of $\frac{1}{L_{up}(z_{srf})-z}$, because the former leads to instabilities in some cases.



The selected formulation for $\delta_{\mathrm{dry}}$ is then:

$$315 \quad \delta_{\mathrm{dry}} = \mathrm{Max}\left(\frac{C_{L_{\mathrm{up}}}}{L_{\mathrm{up}}(z)}, -C_\delta \frac{B_u}{\overline{w_u^2}}\right), \quad \int_z^{z+L_{\mathrm{up}}(z)} \frac{g}{\theta_{v_{\mathrm{ref}}}}\left(\theta_v(z) - \theta_v(z')\right)dz' = -e(z) \tag{16}$$

With this formulation, we assume an updraft parcel is ascending in a nearing environment. Therefore, the updraft virtual temperature, $\overline{\theta}_{v_{\mathrm{up}}}$, is used instead of $\overline{\theta}_v$, at level z, in the upward BL89 calculation. Additionally, we assume that the initial kinetic energy of the parcel is equal to the total grid TKE, as most of the energy originates from non-local structures in convective regimes.

320 The wet part is further complicated to model. Paluch (1979) has shown that entrainment and detrainment rates depend on updraft air mixing with surrounding environment, which triggers evaporative cooling parcels near the cloud edges. This leads to peripheral sinking shells where negatively and positively buoyant parcels coexist locally. We have retained the buoyancy sorting approach in AROME for the cloud layer. The linear interpolation described in Eq. (6) has been replaced by an analytical solution to compute the critical environment fraction, $\chi_c$, derived by de Rooy and Siebesma (2008):

$$325 \quad \begin{cases} \chi_c = \frac{\Delta\theta_v}{\beta\Delta\theta_{il}+(\beta-\alpha)\frac{L}{c_{pm}\Pi}\Delta r_t} \\ \Delta\phi = (\overline{\phi}_u - \overline{\phi}_e) \end{cases} \tag{17}$$

See the reference for the definition of $\alpha$ and $\beta$. The original buoyancy sorting model also assumes a constant fractional mixing rate, $C_{w_{\epsilon+\delta}}$, as shown in Eq. (5). Following Lappen and Randall (2001b), we estimate the updraft dissipation rate, $\epsilon + \delta$, seen as a variable mixing rate between updraft and environment, based on a length scale. Using the same analogy as in the dry part of the ABL, the BL89 length ($L_B$) is suitable to approximate the size of non-local vertical eddies and thus the boundary layer 330 depth:

$$(\epsilon + \delta)_w = \frac{C_{w_{(\epsilon+\delta)}}}{L_B} \tag{18}$$

$L_B = 2\frac{L_{\mathrm{up}}L_{\mathrm{down}}}{L_{\mathrm{up}}+L_{\mathrm{down}}}$ is the Bougeault-Lacarrère length, $L_{\mathrm{up}}$ is the upper part defined in Eq. (16) and the downward part $L_{\mathrm{down}}$ is defined as $\int_{z-L_{\mathrm{donw}}}^z \frac{g}{\theta_{v_{\mathrm{ref}}}}\left(\theta(z') - \theta(z)\right)dz' = -e(z)$. The subscript '$w$' refers to the 'wet' part of the ABL. Figure 3 shows the effects of modified entrainment and detrainment for dry and wet parts, with $C_{w_{(\epsilon+\delta)}} = 0.4$.





**Figure 3.** Temporal evolution of the fractional entrainment rate $\epsilon$ $[\text{km}^{-1}]$ (first and third rows) and the fractional detrainment rate $\delta$ $[\text{km}^{-1}]$ (second and fourth rows) for the LES conditional sampling (first column), the CTRL experiment (second column) and the CTRL + modified $\delta/\epsilon$ experiment (third column). The first two lines correspond to the FIRE case, and the last two to the ARMCu case.

335   For the LES, fractional entrainment and detrainment are calculated using the Couvreux et al. (2009) conditional sampling, with the help of Eq.(3). For the ARMCu case, $\delta$ discontinuities are removed at 13:00 UTC, while the diurnal cycle of the wet part is improved for both the entrainment and detrainment rates in comparison to the CTRL experiment. However, entrainment seems slightly underestimated in the cloud. The result is less clear for the FIRE case, as the sum of the entrainment and



detrainment rates is overestimated, even though the entrainment of the cloud top is improved for the whole simulation. It seems that compensating subsidence of dry air into the ABL deteriorates both quantities in the stratocumulus cloud. In both cases, the modified formulations appear to slightly improve the representation of both quantities.

## 3.3 Turbulence scheme

### 3.3.1 Consistent TKE budget

The eddy diffusive contribution of the EDMF is handled by the CBR scheme. The fluxes and variances of moments, non-precipitating water and liquid potential temperature over the entire horizontal grid box are diagnosed on the basis of K-gradient formulations and the TKE prognostic equation ((2)). The CBR scheme is designed for homogeneous isotropic turbulence. The mass flux part of the EDMF framework can diagnose the anisotropic turbulence quantities of PMMC09, i.e. all the $\overline{\phi'\psi'}_{\mathrm{MF}}$. The buoyancy flux, $\overline{w'\theta'_v}_{\mathrm{MF}}$, is already included in the grid averaged balance of TKE in the original version of PMMC09. We then added the $\overline{w'u'_h}_{\mathrm{MF}}$ missing fluxes to the TKE equation in the shear production/sinking term:

$$\overline{w'u'_h}\frac{\partial \overline{u}_h}{\partial z} = (\overline{w'_e u'_e} + \overline{w'u'}_{\mathrm{MF}})\partial_z \overline{u} + (\overline{w'_e v'_e} + \overline{w'v'}_{\mathrm{MF}})\partial_z \overline{v} \tag{19}$$

Figure 4 shows the effect of the mass flux contribution on the TKE budget term, $(\overline{w'u'_h})\partial_z \overline{u}_h$, in the ARMCu case, where the initial horizontal velocity profile is uniform $((u,v)=(10,0))$. Consequently, it demonstrates that the MF component can satisfactorily represent the missing flux in accordance with the LES, in contrast to the CTRL experience.

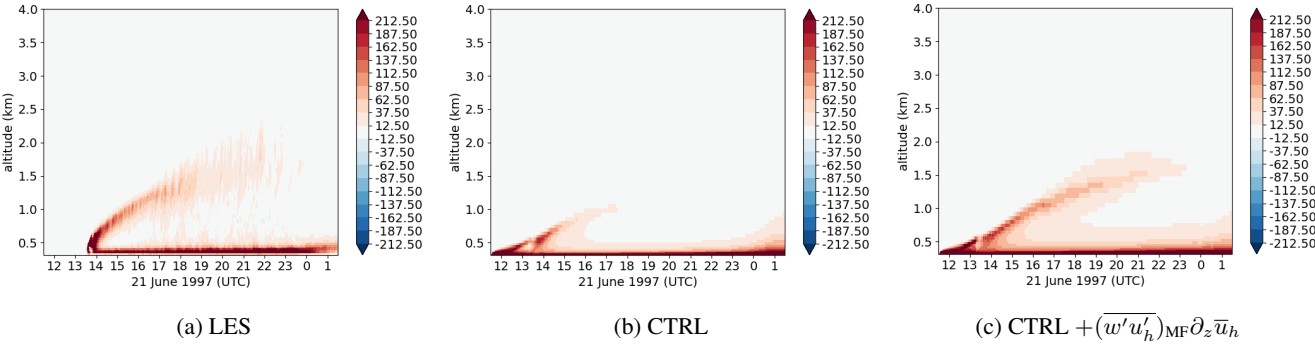

|            (a) LES            |            (b) CTRL            |     (c) CTRL $+(\overline{w'u'_h})_{\mathrm{MF}}\partial_z \overline{u}_h$     |

**Figure 4.** Temporal evolution of $(\overline{w'u'_h})\partial_z \overline{u}_h$ $[m^2 s^{-3}.10^5]$ for the ARMCu case: (a) the LES reference, (b) the CTRL experience and (c) the CTRL + $(\overline{w'u'_h})_{\mathrm{MF}}\partial_z \overline{u}_h$.

More generally, when using EDMF, we should adapt the corresponding equation ((2)), including all the anisotropic contributions provided by the MF part. The EDMF decomposition implies on the TKE:

$$e = (1-a_u)e_e + a_u e_u + \frac{1}{2}\frac{a_u}{1-a_u}\sum_{i=1,2,3}(\overline{u_i}_u - \overline{u_i})^2 \tag{20}$$

 

Thus, the ED turbulent fluxes should be diagnosed using environmental TKE ($e_e$) instead of total TKE ($e$) to avoid double counting, which is not currently done in practice. We must also include an adapted TKE turbulent flux $\frac{1}{2}\sum_i \overline{w'u_i'^2}$.

After some algebra, this leads to:

$$\frac{1}{2}\sum_i \overline{w'u_i'^2} = \sum_j \sum_i a_j \left( \frac{1}{2}\overline{w_j'u_{i,j}'^2} + (\overline{u_{ij}} - \overline{u_i})\overline{w_j'u_{i,j}'} + \frac{1}{2}(\overline{w_j} - \overline{w})\overline{u_{i,j}'^2} + \frac{1}{2}(\overline{w_j} - \overline{w})(\overline{u_{ij}} - \overline{u_i})^2 \right) \tag{21}$$

As in Perrot et al. (2025), Tan et al. (2018) and Witek et al. (2011). Considering $\frac{1}{2}\left(a_u\overline{w_u'u_{i,u}'^2} + (1-a_u)\overline{w_e'u_{i,e}'^2}\right) \simeq \frac{1}{2}\overline{w_e'u_{i,e}'^2}$ and $\frac{1}{2}\overline{w_e'u_{i,e}'^2}$ is performed using the K-diffusion CBR formulation, neglecting $(\overline{u_{ij}} - \overline{u_i})\overline{w_j'u_{i,j}'}$ and rewriting the last terms, we get:

$$\frac{1}{2}\sum_i \overline{w'u_i'^2} \simeq -K_e\frac{\partial e}{\partial z} + \overline{w}_u\frac{a_u}{1-a_u}\left( e_u - e + \frac{1}{2}\sum_i (\overline{u_{iu}} - \overline{u_i})^2 \right) \tag{22}$$

Ultimately, we require a parameterization for $e_u$. Using the prognostic formulation for updraft properties, as in Tan et al. (2018) and Perrot et al. (2025), a simplified formulation of the horizontally averaged updraft TKE equation is:

$$\partial_z e_u \simeq -\frac{\epsilon}{1-a_u}\left( e_u - e - \frac{1}{2}\left(1 + \frac{a_u^2}{1-a_u}\right)\sum (\overline{u_{iu}} - \overline{u_i})^2 \right) \tag{23}$$

This is the same equation as in Perrot et al. (2025).

The last term of Eq. (22) is not negligible, as $\overline{w}_u\frac{a_u}{1-a_u}\sum_i (\overline{u_{iu}} - \overline{u_i})^2 \simeq \overline{w}_u^3$. This has been evaluated on cumulus clouds, where we would expect $\overline{w}_u$ to be the largest in our cases. This significantly improves the TKE evolution, as $\overline{w}_u \simeq 3\mathrm{m.s}^{-1}$ in cumulus plumes. Figure 5 shows TKE transport diagnostics for the term $\frac{1}{2}\sum_i \overline{w'u_i'^2}$. Subfigure (c) shows that adding the total TKE flux Eq. (22) to the model significantly improves the flux distribution compared to LES. However, as updraft vertical velocity, $\overline{w}_u$, is fully diagnosed with PMMC09, inaccuracies and strong temporal variations in its evaluation could cause problems when computing the horizontal mean TKE tendency. Overestimating the value of the updraft vertical velocity can move all the TKE from one level to another, leading to incorrect diffusion mixing in the turbulence scheme. Additionally, the term associated with $\overline{w}_u$ cannot be resolved implicitly with the current schemes; it must be considered as a forcing term. To avoid these issues, we have removed all $\sum(\overline{u_{iu}} - \overline{u_i})^2$ terms from the total flux. Further work in AROME could be carried out to ensure that this term is properly taken into account. Finally, this leads to:

$$\begin{cases} \frac{1}{2}\overline{w'u'^2} & \simeq -K\frac{\partial e}{\partial z} + \overline{w}_u\frac{a_u}{1-a_u}(e_u - e) \\ \partial_z e_u & = -\frac{\epsilon}{1-a_u}(e_u - e) \end{cases} \tag{24}$$

Therefore, we assume that the horizontal grid-averaged TKE is transported passively by the plume.







**Figure 5.** Temporal evolution of $\frac{1}{2}\overline{w'u_i'^2}\;[m^3 s^{-3}]$ for the ARMCu case: (a) LES, (b) CTRL experiment, (c) CTRL experiment with the TKE flux expressed by Eq. (22) and (d) CTRL experiment with the TKE flux of Eq. (24). Note that $a_u << 1$ is still considered here.

Figure 5(d) shows that, while passive turbulent transport of the TKE (Eq. (24)) improves the flux distribution in the simulation, it seems to be underestimated near the ground, where the ED component is negative, i.e., where $\overline{w}_u$ is relatively small in comparison to turbulent diffusion. In conclusion, a compromise is reached by considering Eq. (24), which increases the TKE without the constraints of Eq. (22).





### 3.3.2  Turbulent mixing length

We have also tested another implementation of the turbulent mixing length. A new formulation from Rodier et al. (2017) has been implemented and tested. This is the BL89 mixing length which has been modified to take into account the vertical wind shear effects, and is currently running in the Meso-NH model. This mixing length was primarily designed for very stable conditions, but we expect it to improve our results on sheared updrafts. Finally, we implemented the "ADAP" mixing length from Honnert et al. (2021), which adds a grid-size-dependent limit to the previous mixing length:

$$L_{ADAP} = \text{Min}(\alpha_{\text{turb}} L_\Delta, L_{RM_{17}}) \tag{25}$$

$\alpha_{\text{turb}}$ is a constant, $L_\Delta = (\Delta_x \Delta_y)^{\frac{1}{2}}$ is a geometric length ($\Delta_x$ and $\Delta_y$ being the horizontal dimensions of the grid) and $L_{RM_{17}}$ is the Rodier et al. (2017) length.

### 3.4  Saturation adjustment scheme and rain fraction

Representing subgrid clouds is an important issue in the present parameterization. The CB02 formulation is very close to an unimodal normal statistical distribution, which is typically used to represent homogeneous isotropic turbulence. In unstable ABLs, saturated updrafts are the main source of subgrid condensates, $r_c$. The anisotropy contribution from shallow convection is computed using PMMC09 diagnostics, which adds an extra cloud fraction and condensates to the CB02 formulation. While the schemes produce subgrid clouds satisfactorily, they do not respect the grid decomposition, as CB02 is computed over the entire grid area. Additionally, many studies, such as Perraud et al. (2011), indicate asymmetrical distribution of conservative variables for ABLs. With these considerations, we decided to implement a statistical PDF-based scheme instead of the CB02 and PMMC09 cloud schemes. This approach enables us to consider all cloud properties from a single distribution. Consistently with turbulence diagnostics, we retained the saturation deficit $s$ as the PDF variable. Note that, in most cases of shallow convection, choosing either $r_t$ or $s$ may not be important as it will produce identical results, since we expect updrafts to be fully saturated (Perraud et al., 2011) (cumulus clouds for example). The EDMF decomposition on the whole horizontal grid box for $s$ is then:

$$\begin{cases} \overline{s} = a_u \overline{s}_u + (1 - a_u) \overline{s}_e \\ \text{PDF}(s) = C_{cf} a_u \text{PDF}_u(s) + (1 - C_{cf} a_u) \text{PDF}_e(s) \end{cases} \tag{26}$$

$\text{PDF}_u$ and $\text{PDF}_e$ are unknown Probability Density Functions and $C_{cf}$ is a parameter. Several PDF expressions have been proposed in recent decades. According with the CBR scheme, we would expect the $\text{PDF}_e$ to be close to a normal distribution centred on the environmental saturation deficit, $\overline{s}_e$, although this is not obvious. The shape of $\text{PDF}_u$ is even less clear. For simplicity, another normal distribution is used. These choices were made thanks to the work of Jam et al. (2012) and Perraud et al. (2011). The chosen distributions are then:

$$\text{PDF}_i(s) = \frac{1}{\sqrt{2\pi} \overline{\sigma}_i} e^{-\frac{1}{2}\left(\frac{s - \overline{s}_i}{\overline{\sigma}_i}\right)^2} \tag{27}$$




Thus, the cloud fraction $CF$ and the cloud condensate mixing ratio $\overline{r_c}$ are:

$$
\begin{cases}
CF = \int_0^{+\infty} \mathrm{PDF}(s)ds \\
\overline{r_c} = \int_0^{+\infty} s\mathrm{PDF}(s)ds
\end{cases}
\tag{28}
$$

Using a PDF enables a two-part decomposition of the cloud, as described in Turner et al. (2012) (see Fig. 6). This approach maintains complete consistency with the cloud scheme without introducing a new PDF. Consequently, the subgrid cloud fraction and mixing ratio become:

$$
\begin{cases}
CF = CF_H + CF_L \\
\overline{r_c} = \overline{r_{cH}} + \overline{r_{cL}}
\end{cases}
\tag{29}
$$

The subscript **H** refers to the part of the cloud with a content above the ICE3 autoconversion threshold, while the subscript **L** refers to the part below. From the previously diagnosed PDF, we can easily compute $CF_H$ and $\overline{r_{cH}}$:

$$
\begin{cases}
CF_H = \int_{\frac{\rho c_{\mathrm{crit}}}{\rho_{\mathrm{ref}}}}^{+\infty} \mathrm{PDF}(t)dt \\
\overline{r_{cH}} = \int_{\frac{\rho c_{\mathrm{crit}}}{\rho_{\mathrm{ref}}}}^{+\infty} s\mathrm{PDF}(t)dt
\end{cases}
\tag{30}
$$

The formulation enables us to retrieve the cloud fraction of updrafts whether or not they are precipitating, and also to preserve both the cloud fraction and its heterogeneity for the autoconversion process, whether or not the clouds are driven by updrafts.

Additionally, the evaporation process has been modified to take into account the rain fraction $CF_{\mathrm{rain}}$, following Turner et al. (2012). Figure 6 resumes the new subgrid cloud scheme implemented in AROME and its interaction with the microphysics scheme. The leftmost normal mode is based on the environment characteristics, and the rightmost one is diagnosed from PMMC09. Note that $\mathrm{PDF}_e$ is calculated from grid box-averaged properties, meaning it is too far to the right ($\overline{s} > s_e$) and too wide ($\overline{\sigma} > \overline{\sigma}_e$). Figure 6b shows the rain fraction $CF_{\mathrm{rain}}$ being deducted such that, at level $J$, $CF_{\mathrm{rain}}^J = \mathrm{Max}(CF_{\mathrm{rain}}^k, CF_H^k)$, $J_{\mathrm{top}} \geq k \geq J$, where $J_{\mathrm{top}}$ is the top model level. This enables ICE3 evaporation to occur on $CF_{\mathrm{rain}}$ rather than on the entire grid.





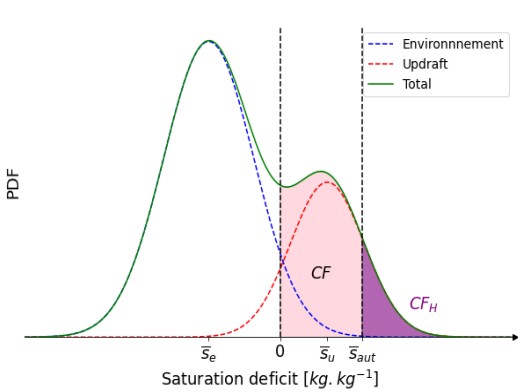
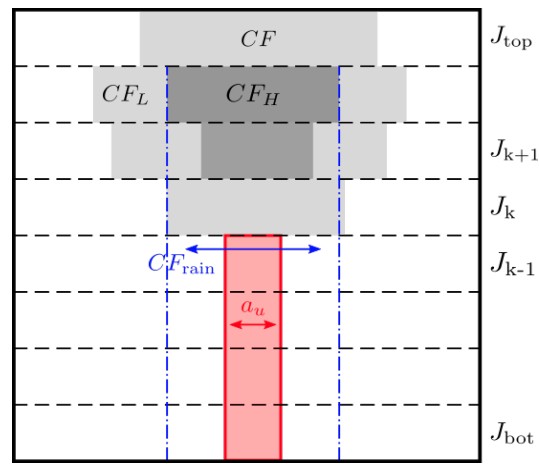

(a) Subgrid cloud bi-gaussian scheme implemented in the new AROME version: environmental normal mode $\{\bar{s}_e, \overline{\sigma_{s_e}}\}$ (blue curve), normal updraft mode $\{\bar{s}_u, \overline{\sigma_{s_u}}\}$ (red curve) and the total distribution (green curve). $CF$ and $\overline{r_c}$ are diagnosed from the PDF for $s \geq 0$ and $CF_H, \overline{r_{cH}}$ for $s \geq \frac{q_{c_{crit}}}{\rho_{ref}}$.

(b) Precipitation scheme in the new AROME version. $CF_H$ is the dark grey area, $CF_L$ is the grey area, the red area is the updraft fraction and the dashed blue lines delimit the $CF_{rain}$ area. The top model level is referenced as $J_{top}$ and the bottom model level is $J_{bot}$.

**Figure 6.** Changes summary on the diagnostic cloud and the microphysical scheme.

Finally, a bi-normal PDF system requires closures for both $\overline{\sigma_s}$. In two-subgrid interactive objects such as EDMF, most of the variance budget comes from exchanges between the updraft and the environment, i.e. entrainment and detrainment. See the prognostic equation for subdomain variance in Tan et al. (2018) and Cohen et al. (2020). While these equations could connect EDMF with the cloud scheme, they require further closures and careful implementation in the involved schemes. In this work, we retain a parameterization to overcome these constraints. Various sets of parameterizations in the literature take into account the interaction between the environment and the updraft using the quadratic difference of the saturation deficit $(\bar{s}_u - \bar{s}_e)^2$ (for example, Jam et al. (2012) and Perraud et al. (2011), not exhaustive). We have therefore adopted a very simple empirical formulation:

$$
\begin{cases}
\overline{s_e'^2} = \underbrace{\overline{s_{ED}'^2}}_{CBR} + C_{\sigma_e}(1 - a_u)(\bar{s}_e - \bar{s})^2 \\
\overline{s_u'^2} = C_{\sigma_u} a_u (\bar{s}_u - \bar{s})^2
\end{cases}
\tag{31}
$$

Where $C_{\sigma_e}$ and $C_{\sigma_u}$ are parameters. In Eq. (31), we simply assume that the internal variances of the subdomains are proportional to the structural terms of Eq. (11). Figure 7 shows cloud diagnostics when considering of a Bi-Normal (BN) PDF only, with $C_{cf} = 2.75$, $C_{\sigma_e} = C_{\sigma_u} = 1.0$. In the BN experiment, we also use Eq. (17) for $\chi_c$ to prevent cloud instabilities.





**Figure 7.** Implementation of a Bi-Normale (BN) PDF scheme without modified autoconversions. Time evolution of the vertical profiles of cloud fraction for the FIRE case: (a) CTRL experiment and (b) BN experiment. Temporally averaged vertical profiles of (c) cloud fraction and (d) liquid water content for the RICO case between 20:00 UTC and 24:00 UTC.

For the RICO case, the cloud fraction at the base is highly reduced by 20%, and the liquid cloud content by 15%, bringing the 1D simulation closer to the LES. The FIRE case shows slight improvement at the cloud base and top, but the cloud remains 445 too close to the ground.





## 4 HighTune Explorer experience

### 4.1 Metrics and parameters selection

A brief description of the HTexplo statistical tool is provided in Section 2.3 and in appendix A. In this section, we set up the HTexplo experience to explore the modified AROME version. As previously explained, history matching requires an initial
parameter space $\text{NROY}^0$ and metrics definitions in order to statistically capture the underlying physics of the ABL. Section 2.1 describes the use cases for AROME development. These fairly represent the range of convective cases encountered in the ABL. The choice of metrics is important yet subjective; it must enable simulations to approach the reference simulations without hindering the tool's convergence. We have chosen simple metrics that best represent the cloud layers, which are the end product of the subgrid physical schemes. Consequently, many of them are calculated directly on the cloud representation:

➤ **P-**$\theta = \frac{\int_0^{z_{\max}} \text{Min}((\theta(z,t)-\theta(z,t=1)),0)dz}{\int_0^{z_{\max}} dz}$ is a deviation from the initial $\theta$ profile, which is a proxy for the boundary layer top entrainment.

➤ $\overline{\text{X}}_{z_{\min}}^{z_{\max}}$ is a vertical average of the variable X (which may be TKE or $\theta$) between $z_{\min}$ and $z_{\max}$.

➤ $\overline{\text{CF}}^z = \frac{\int_0^{z_{\max}} CF(z)zdz}{\int_0^{z_{\max}} CF(z)dz}$ is an average cloud height definition.

➤ $\overline{\text{Max(CF)}}^z = \frac{\int_0^{z_{\max}} CF(z)^4 zdz}{\int_0^{z_{\max}} CF(z)^4 dz}$ is a cloud average, which can be interpreted as the maximum cloudiness height.

➤ $\text{Z}_{\text{base}}$ is the cloud base height.

➤ $\text{Z}_{\text{top}}$ is the cloud top height.

➤ $\text{CF}_{\text{max}}$ is the maximum cloudiness in the model column.

➤ **LWP** is the Liquid Water Path

All metrics, except $\overline{\text{X}}_{z_{\min}}^{z_{\max}}$ and LWP, are calculated using a time average over a period. Figure 8 summarizes the metrics and calculation periods for each case.

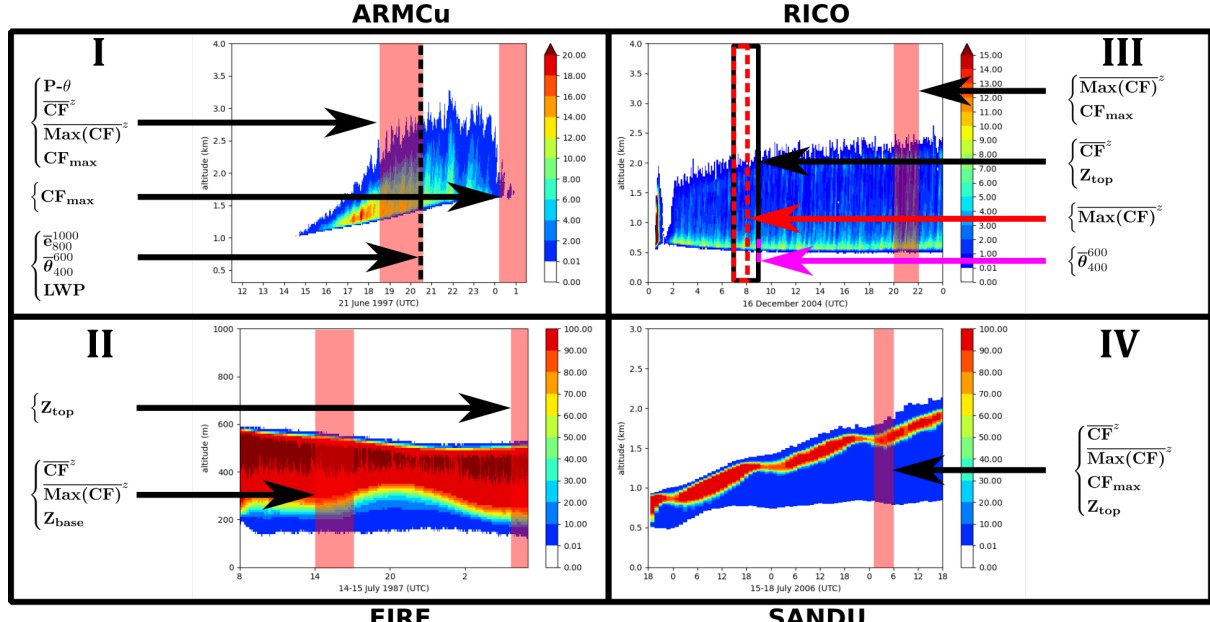

**Figure 8.** Selection of metrics for the HTexplo experience. The time evolution of the vertical profiles of the cloud fraction is shown for the reference LES of ARMCu (top left, I), FIRE (bottom left, II), RICO (top right, III) and SANDU (bottom right, IV) cases. The red shadings and boxes correspond to the areas where the metrics are computed for each case, colored arrows link the specific metrics to their corresponding time averages (see the text for more details).

SCM simulations can be run between waves and compared with LES to provide more detailed diagnoses. These diagnoses can help identify any shortcomings in the parameterizations used. This evaluation provides insight into model biases, and ensures
that HTexplo converges to a parameter space that is physically representative of the ABL.

For the ARMCu case, a first set of metrics (P-$\theta$, $\overline{CF}^z$, $\overline{Max(CF)}^z$, $CF_{max}$) is averaged between 18:30 UTC and 20:30 UTC. This corresponds to a mature phase of cumulus clouds, when surface sensible and latent fluxes are at their maximum. Vertically averaged values ($\overline{e}_{800}^{1000}$, $\overline{\theta}_{400}^{600}$ and LWP) were also added at 20:30 UTC to ensure consistency between the SCM and LES for the TKE, potential temperature profile, and cloud liquid water content. Additionally, $CF_{max}$ is added at 00:00 UTC to prevent
cloud formation at the end of the simulation.

For the FIRE case, we aimed to eliminate cloud biases for stratocumulus, such as boundary layer oscillations and cloud collapse. Thus, the first temporal period (from 14:00 UTC to 17:00 UTC) focuses on nocturnal stratocumulus, measuring the height and shape of the cloud ($\overline{Max(CF)}^z$, $\overline{CF}^z$, $Z_{base}$). Stratocumulus oscillations may be linked to ABL deepening too far into the very dry stable layer. To address this, the $Z_{top}$ metric was added to calculate the cloud top at the end of the diurnal cycle (between
06:00 UTC and 08:00 UTC).

The RICO case is almost identical to the ARMCu case. Thus, similar metrics ($\overline{CF}^z$, $Z_{top}$ and $\overline{Max(CF)}^z$) are extracted in the mature phase of the cumulus cloud (between 07:00 UTC and 09:00 UTC) and at the end of the simulation ($\overline{Max(CF)}^z$ and





$CF_{max}$ between 20:00 UTC and 22:00 UTC). Additionally, for the same reason as for the ARMCu case, the metric $\overline{\theta}_{400}^{600}$ is added here.

The last case is the stratocumulus transition SANDU, for which we want to ensure ABL deepening due to progressively rising marine surface forcings (SST in this case). Therefore, we use the metrics $\overline{Max(CF)}^z$, $\overline{CF}^z$, $Z_{top}$ and $CF_{max}$ on 18 July from 01:00 UTC to 06:00 UTC, at the end of the simulation.

Besides, AROME schemes contain many free parameters. Not all of these can be considered, as this would require a very large number of SCM simulations to correctly sample the space, which would be very costly. Instead, we focus on the parameters

that are the most significant for shallow convection. This includes 10 parameters for the turbulence, convection, and saturation adjustment schemes. Table 1 summarizes these parameters and the $NROY^0$ starting space.

| AROME parameters | | Parameter space $NROY^0$ | |
|---|---|---|---|
| Parameter | Equation | Min value | Max value |
| $C_{w_{\epsilon+\delta}}$ | Eq: (18) | 0.0 | 0.6 |
| $b'$ | Eq: (15) | 0.0 | 1.0 |
| $C_{\epsilon_{dry}}$ | Eq: (4) | 0.1 | 0.55 |
| $a$ | Eq: (15) | 0.6 | 1 |
| $C_{\sigma_u}$ | Eq: (31) | 0.0 | 10.0 |
| $C_{\sigma_e}$ | Eq: (31) | 0.0 | 10.0 |
| $C_{diss}$ | Eq: (2) | 0.1 | 0.85 |
| $C_{\delta}$ | Eq: (16) | 0.0 | 10.0 |
| $C_{cf}$ | Eq: (26) | 0.9 | 2.75 |
| $\alpha_a$ | Eq: (15) | 0.0 | 0.2 |

**Table 1.** Setting parameters for HTexplo experiences.

The range of parameter values for $NROY^0$ is based on documented values from other studies (Gregory, 2001; Pergaud et al., 2009; de Rooy and Siebesma, 2010; Rio et al., 2010; Jam et al., 2012; Rodier et al., 2017; Tan et al., 2018). We have chosen to run 10 times as many SCM simulations as there are free parameters, i.e. 100 MUSC simulations per reference case and per

HTexplo wave. This is a common approach in the parameterization community for the Htexplo configuration (Williamson et al., 2017; Hourdin et al., 2021; Audouin et al., 2021). Once the emulators are set up, the parameter space can be resampled with any number of points. Ideally, we would provide as many points as possible. However, in practice, because we use many metrics for the ABL cases and parameters, emulating the parameter space too finely quickly leads to excessive computation time with the current tool version. Instead, we resample the space with an order of $10^6$ points, bringing us down to approximately 4 emulator

points per dimension in the parameter space. This provides a sufficient global overview of the final NROY. The risk is to obtain a noisy NROY and miss certain plausible regions of parameter values in the 10-dimensional space.



The metrics for each case are added progressively as the HTexplo waves progress, enabling the impact of each reference case to be analyzed in the parameter space. Table 2 summarizes the general design of the HTexplo experience.

| Cases | Evaluated Metrics | Wave | | | |
|-------|-------------------|------|------|------|------|
| **ARMCu** | Figure 8: **I** | 1 | 2 | 3 | ≥ 4 |
| **FIRE** | Figure 8: **II** | | | | |
| **SANDU** | Figure 8: **IV** | | | | |
| **RICO** | Figure 8: **III** | | | | |
| | | $(T, \tau)$ | | | |
| | | (3,0) | (3,1) | (3,1) | (3,1) |

**Table 2.** HTexplo experience general design. The first column corresponds to the cases used in HTexplo, and the second column to their selected metrics. The last columns show the wave to which each case is added. The bottom of these columns show the chosen tolerance T and the number of metrics $\tau$ allowed to be far for the reference (see appendix A), for each wave.

The cases are added in the following order: ARMCu, FIRE, SANDU and RICO. For all waves, the cutoff is set to the default value of 3. However, a metric rejection ($\tau = 1$, see appendix A) is possible from the introduction of the FIRE case, as some metrics may be too restrictive. After wave 4, further iterations are performed until the remaining space is stabilized, i.e., when the uncertainty of the emulators is similar to the internal uncertainty of the model or the uncertainty of the LES metrics.

## 4.2 HighTune Explorer results

Figure 9 shows a visualization of the parameter space at the end of wave 4, after all the metrics have been taken into account for the first time in the experiment.





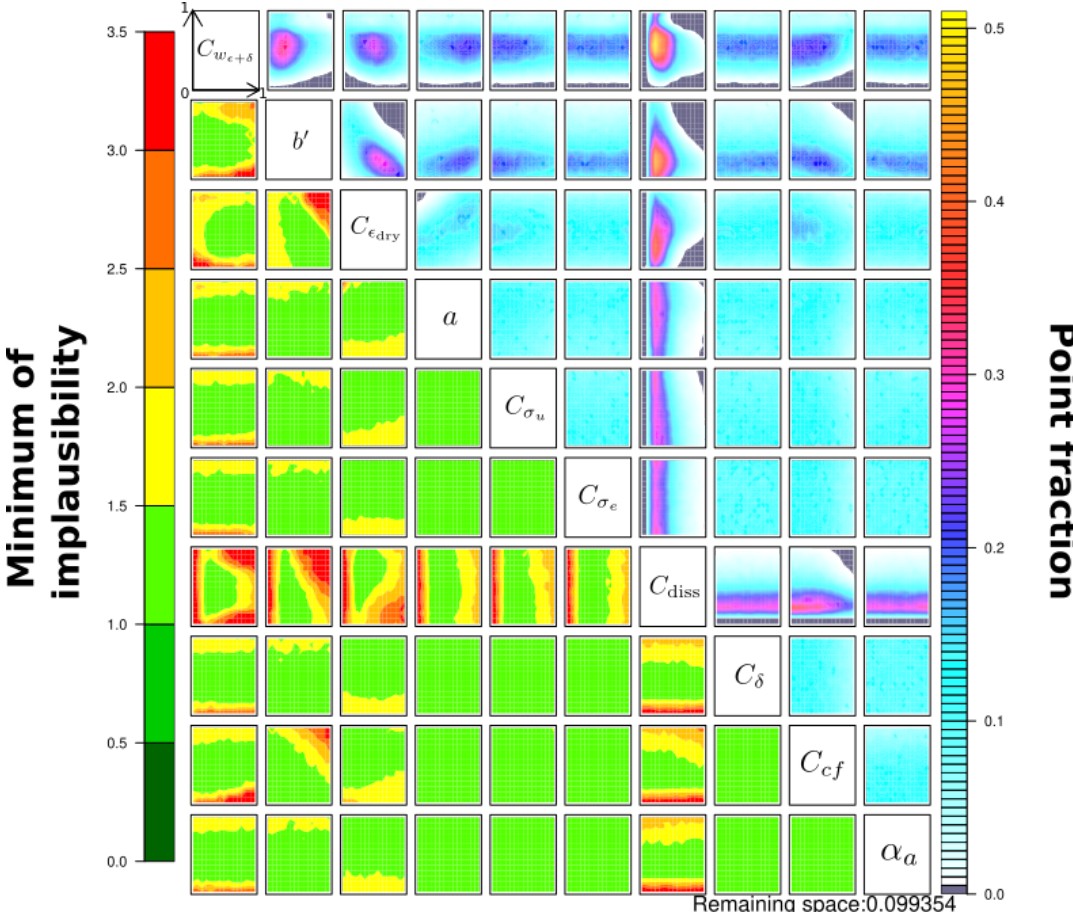

**Figure 9.** Implausibility matrices for NROY[4] using the 10 parameters selected in Table 1 and the wave design referenced in Table 2. The lower left triangle section is the minimum implausibility found by HTexplo and the upper right triangle section is the fraction of the emulated point respecting the tolerance. The full description of this figure is given in the text.

In accordance with Couvreux et al. (2021), the upper right triangle contains multiple matrices (in shades of blue), each of which is a restriction to two parameters of the parameter space. These 2 parameters are shown on the main diagonal in boxes of the same row and column. The parameter values are normalized between 0 and 1 (minimum and maximum range values). Each matrix shows the fraction of emulated points in agreement with the metrics, when taking all waves into account, and all

remaining parameters varying randomly. Some matrices show anisotropy and are highly dependent on the value of the parameter set. For instance, the first colored matrix (in the first row and second column) illustrates the proportion of emulated points for fixed $C_{w_{\epsilon+\delta}}$ relative to $b'$. In this case, it shows that weak values of $b'$ and medium values of $C_{w_{\epsilon+\delta}}$ are preferably retained given the chosen tolerance. The grey regions represent all NROY[0] points that have been removed in successive waves. The lower left triangle (in shades of green) is similar to the upper right one, but presents the minimum value of the implausibility

found by HTexplo instead. Using the same parameters as in the previous example, the colored matrix in the second row and





first column indicates lower implausibility for these values, meaning better representation of the selected metrics by SCM.

Unlike the example of $C_{w_{\epsilon+\delta}}$ relative to $b'$, the parameters $C_{\sigma_e}$, $\alpha_a$ and $C_\delta$ seem to have a small influence on the metrics, even though they impact the final MUSC simulations. This suggests that the metrics may be inappropriate for tuning these parame-

ters. It turns out that $C_{w_{(\delta+\epsilon)}}$, $b'$, $C_{\epsilon_{\text{dry}}}$ and $C_{\text{diss}}$ are consistently the most important parameters of the current parameterization. Note that there may be correlations between some parameters, as they are not completely independent. An example of this is the link between $b'$ and $C_{\epsilon_{\text{dry}}}$, which is clearly visible in Fig. 9. In this case, the relations given in Eq. (4) and Eq. (15) can be used to show that a small $b'$ leads to a larger $\overline{w}_u$. This means that $C_{\epsilon_{\text{dry}}}$ must also be larger to achieve the correct value of $\epsilon_{\text{dry}}$. In general, however, the relations are more complicated.

After a dozen successive waves, less than $2\%$ of NROY$^0$ remains. Adding more waves is not relevant, as this only reduces the parameter space by $0.01\%$. The reduced parameter space enables us to manually select a set of plausible parameters. However, there is still uncertainty regarding the parameters that are less sensitive to the chosen metrics, such as $C_{\sigma_u}$, $C_{\sigma_e}$ and $C_\delta$. The final selected set is as follows: $\{C_{w_{\epsilon+\delta}} = 0.34,\ b' = 0.13, C_{\epsilon_{\text{dry}}} = 0.35,\ a = 0.67,\ C_{\sigma_u} = 2.00,\ C_{\sigma_e} = 5.08,\ C_{\text{diss}} = 0.30,\ C_\delta = 9.60,\ C_{cf} = 1.90, \alpha_a = 0.05\}$, and will be referred to as the 'NEW' version of AROME.

## 4.3    Tuned MUSC results

This subsection illustrates the modifications set out in section 3 for the configuration of the parameters given in the previous section, as compared with LES.

## 4.4    Cloud fraction

First, we examine the evolution of the cloud cover (Fig. 10), which reflects the structure of the underlying plumes.





**Figure 10.** Time evolution of the vertical profiles of cloud fraction [%] for ARMCu (first row), FIRE (second row), RICO (third row), and SANDU (last row) cases with the reference LES (first column), the CTRL experiment (second column), and the selected SCM simulation from the HTexplo experiment (third column).

The NEW version of AROME reproduces most cases well. For RICO and ARMCu cumulus, the cloud structure is accurately simulated, with an attenuation of the overestimation of the cloud fraction at the cloud base. Sensitivity tests (not shown) suggest that this behavior is partly due to the separation of the environment and updraft subdomains, introduced by the binormal PDF-




based cloud scheme. This allows for better diagnosis of cloud content and fraction. However, modifications to the cloud scheme alone cannot fully explain the improved stratocumulus transition in the SANDU case. This accounts for the modifications

introduced in the shallow convection and turbulence schemes, which ensure that the turbulent fluxes more accurately represent updrafts. Additionally, the AROME NEW experiment enables all the diurnal cycles to be recovered, particularly the FIRE stratocumulus. This is less evident for the SANDU case, although the transition is quite faithful to the LES. The cloud base for the FIRE case is also much improved, with the cloud no longer reaching the ground. Nevertheless, some biases remain, such as a slight underestimation of the cloud base in the convective cases (mostly evident in the ARMCu case). Most cloud

tops are underestimated too, except for FIRE (it has been shown that the LES is sensitive to the microphysical scheme, with cloud tops fluctuating between 500 and 600 metres). An underestimation of the cloud fraction, particularly in the initial phase of convection (see ARMCu) is also noticeable.

### 4.4.1   Updraft properties

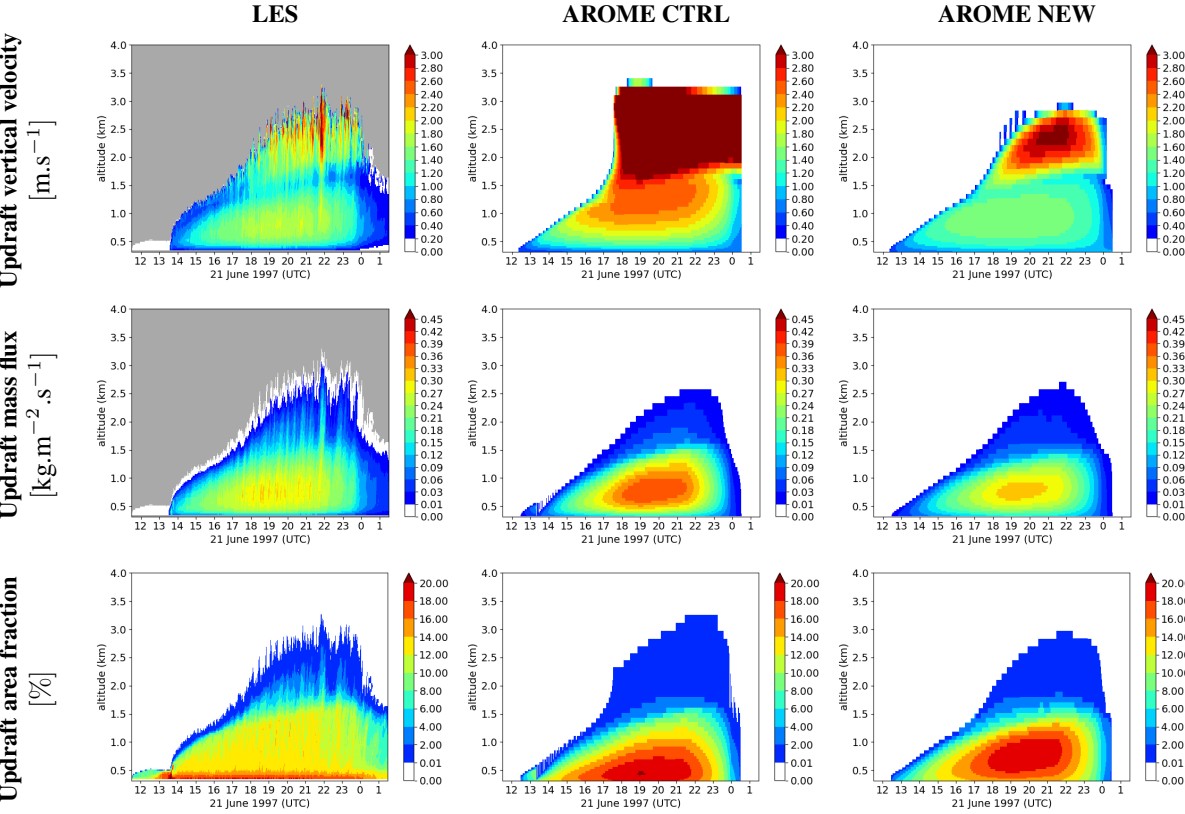

**Figure 11.** Time evolution of the vertical updraft properties for the ARMCu case: updraft vertical velocity $\overline{w}_u$ (first line), updraft mass flux $M_u$ (second line), and updraft fraction $a_u$ (third line) with columns same as Fig. 10. LES diagnostics are obtained with the conditional sampling method.





The cloud representation is closely linked to the mass flux part of the parameterization. Figure 11 shows a comparison of the
LES and MUSC updraft properties for the ARMCu case. While there is still an overestimation of the mean vertical updraft
velocity in the cloud layer, it has improved significantly. The updraft mass flux is also improved in the dry layer, but is
underestimated in the cloud layer. This is consistent with a too weak $\epsilon$ and an overestimated $\delta$ in the cloud for the NEW
experiment compared to the LES (not shown). The combined modifications to $\overline{w}_u$ and $M_u$ are illustrated in the updraft area
fraction variable, which is similar to the LES conditional sampling, but too strong at the bottom of the ABL and too weak
at the top. Consequently, the cloud fraction is underestimated, and the $C_{cf}$ parameter partially corrects the biases for cloud
diagnostics.

### 4.4.2 Rain water content

Figure 12 illustrates precipitation in the NEW AROME experiment (the CTRL configuration produced no rain at all) and shows
an improved representation of the rain water content, $\overline{q_r}$, for all cases. Although precipitation is underestimated in all cases
except ARMCu, its temporal and spatial occurrence is quite good, representing a significant improvement.



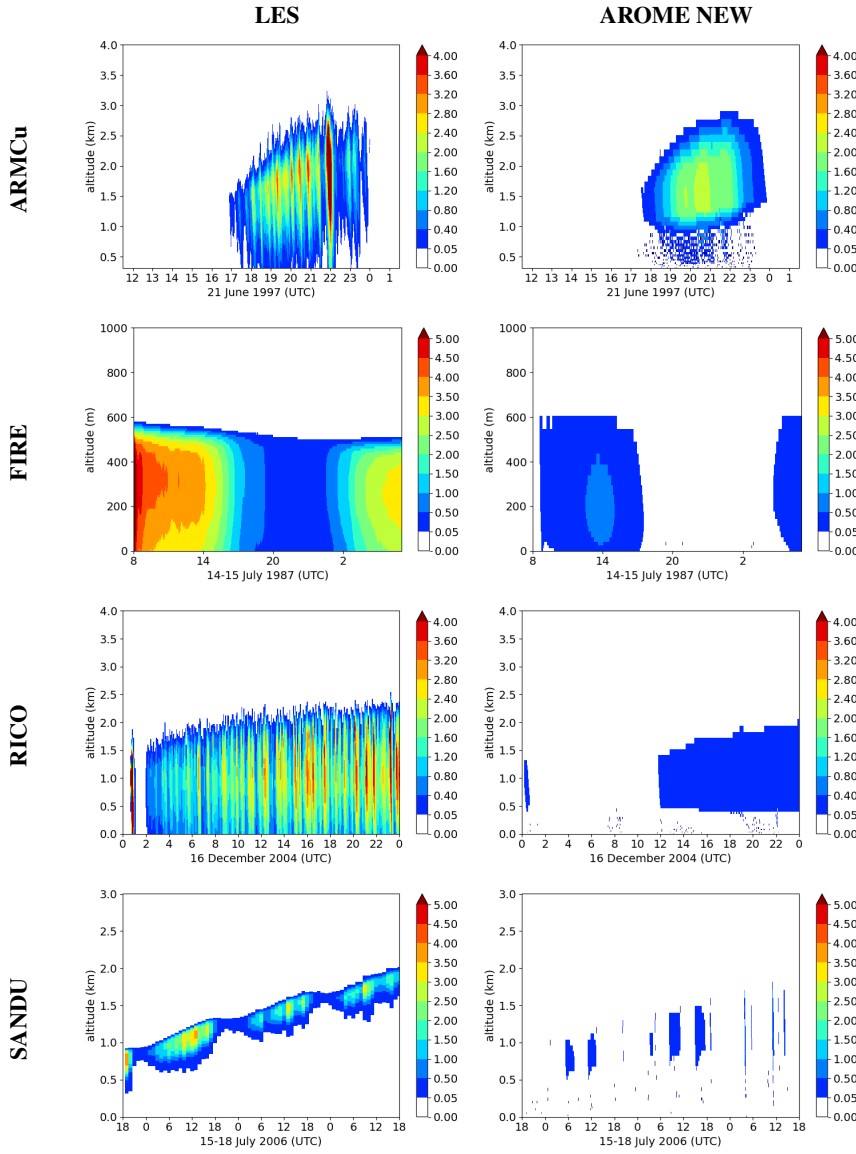

**Figure 12.** Time evolution of the vertical rain water content $\overline{q_r}$ [mg.kg$^{-1}$] for ARMCu (first row), FIRE (second row), RICO (third row), and SANDU (last row) cases with the reference LES (first column) and the NEW AROME experiment (second column). No CTRL experiment is provided, as all rain water contents are equal to zero in the AROME operational configuration.





### 4.4.3 TKE

**Figure 13.** Same as Fig. 10 but for TKE $[m^2 s^{-2}]$ and without the SANDU case.

Figure 13 shows the TKE underestimation in all cases for the CTRL experiment. LES TKE is calculated using the total contribution (resolved plus subgrid) across the entire horizontal domain. For the AROME NEW experiment, TKE is improved for all cases, but further improvement is required to reach LES intensity levels (e.g. the FIRE case). The main TKE biases are 560 present in the cloud layer; however, the dry part of the ABL is in good agreement with LES, except for the surface layer up to 100 m above the ground. For the FIRE case, the TKE diurnal cycle is correctly reproduced, while a strong underestimation is present near the inversion. In this case, missing fluxes in the parameterization, such as the pressure term, appear to be significant (see Appendix B for the TKE source and sink terms of the FIRE case).



## 5 Discussion

In this section, we discuss the uncertainties and questions regarding the parameterization framework of the AROME model. The original AROME version had difficulty agreeing with LES energy diagnostics. In particular, the TKE was far too weak to accurately represent the ED component of the fluxes. This behavior has already been identified for clouds and deep convection (Verrelle et al., 2015), and current work in AROME has attempted to evaluate and implement an anisotropic turbulence parameterization following the work of Moeng (2014) and Leonard (1975). For the ABL, Fig. 5 shows that the $\overline{w'u'^2}_{\mathrm{MF}} \approx \frac{a_u}{1-a_u}\overline{w}_u^3$ missing term of Eq. (22) is an important contribution and should be added, particularly in strong convective regimes such as ARMCu. One way to implement this stably and with temporal consistency would be to introduce a prognostic version of EDMF for $\overline{w}_u$ to carefully close grid-averaged TKE with updraft kinetic energy and avoiding double counting for K-gradient.

It is not clear what causes the TKE bias in Fig. 13, although the production and transport terms are globally better represented in AROME NEW compared to AROME CTRL for the FIRE TKE budget (Fig. B1). The pressure correlation term from the LES suggests that is not negligible at the stratocumulus top (not taken into account in AROME). Indeed, the literature on this term is quite extensive (for example, Canuto et al., 2001). Furthermore, the choice of the dissipation and mixing lengths is also questionable. Currently $L_{\mathrm{BL89}}$ is used for both, but a Deardorff-like local length (Deardorff, 1980) could be used instead. Conceptually, we envisage that the anisotropic component of the turbulence spectrum in the ABL is realised by the shallow convection scheme. However, in AROME, no physics is provided to represent these turbulence structures in the upper troposphere. We did not use local lengths so.

The difficulties in reproducing the upper part of FIRE are probably linked to the absence of relevant structures, such as triggered radiative cooling downdrafts near the inversion and cloud layers. In their study of stratocumulus, Brient et al. (2019) performed a LES analysis of downdrafts in the FIRE case and pointed out non-negligible effects in the ABL top. The biases retrieved from the AROME SCM (especially the overestimation of the mass flux and the underestimation of the TKE at the cloud top) suggest that downdraft modeling should be added. The main difficulties encountered are the initialization and the entrainment/detrainment closures, although schemes with downdrafts have already been implemented in GCMs with simple solutions. Some biases persist in the cloud layer: mostly too weak fluxes, a lower cloud top and base, and too strong vertical velocity. Some of these inaccuracies are due to an underestimation of wet entrainment; thus, the mass distribution PDF from the Bretherton et al. (2004) assumption or weak mixed mass (Eq. (5)) at the cloud level might be the cause.

Although further closures are required, an alternative diagnostic formulation of full variance equations could be implemented instead of an empirical parameterization of internal variances fed into the cloud PDF. Furthermore, the model does not process the shallow convection cloud scheme and the turbulence cloud scheme in the same way (see section 2.2.4). Building a complete PDF enabled us to parameterize the autoconversion process consistently with the cloud, and to produce precipitation where the CTRL experiment did not permit it. However, sensitivity tests revealed that excessive precipitation could also cause the





cloud layer to break up and collapse (not shown). This effect is not negligible, as it can lower the surface temperature by a few Kelvins and create small cold pools, and/or drain the liquid cloud water content too much.

The importance of modifying the evaporation process is unclear in the simulations as it seems to have a negligible effect on sensitivity tests. For the SANDU case, it turns out that precipitation occurring too low in the ABL indicates an underestimation of this process. An overestimation of $q_r$ under the clouds can explain the observed reduction in cloud base, as seen in Fig. 10. Nevertheless, LES should be used with caution for precipitation, as it depends strongly on the microphysical scheme. Comparing with observations could show how accurately AROME predicts rainwater content.


Uncertainties also arise from HTexplo, since the tool requires subjective decisions to be made regarding acceptable tolerances for the scalar metrics. This introduces uncertainty regarding the free model parameters. If we are not sufficiently restrictive (i.e. if we use too few metrics and/or a too large tolerance), there is a risk that we will include parts of NROY[0] that we do not want. Conversely, adding too many metrics or setting the tolerance too low risks producing an empty parameter space. As can be seen

in Figure 9, some parameters are less sensitive to metrics such as $C_{cf}$, $C_{\sigma_e}$, $C_{\sigma_u}$, $\alpha_a$ or $C_\delta$. This does not mean that they are irrelevant for parameterization. Convergence of HTexplo on a plausible parameter space enables us to extract information about parameterization biases from sampled MUSC simulations within that space. However, this does not tell us which closures or parameterizations are questionable or incomplete; further investigation is required. Most values can be physically justified and depend on the convection regime being investigated. Finally, the parameter uncertainties in the simulations could be reduced

by using special reference cases and metrics.

Despite the remaining issues in the model, the EDMF framework of AROME is effective in representing shallow clouds.

## 6  Conclusions

This study investigated the AROME physics in the ABL, focussing on deficiencies in the representation of shallow convection,
particularly the misrepresentation of the stratocumulus diurnal cycle and the lack of precipitation. An update of the AROME parameterization package has been described, which includes improvements and testing of simple formulations in the shallow convection, turbulence, cloud, and microphysics subgrid schemes. Efforts have been made to improve physical consistency between the AROME schemes. These modifications have been numerically tuned by the statistical history-matching tool HTexplo using simple representative metrics on four ABL regimes. Evaluations against reference LES performed with the Meso-NH
research model demonstrate an overall improvement in all ABL regimes, from cumulus to stratocumulus, including transitions. The model can accurately reproduce cloud fractions, cloud water content, and turbulence according to LES conditional sampling diagnostics. Rain water content can also be partially predicted by linking the autoconversion process to the PDF used by the cloud scheme. Corrections to vertical updraft velocity and mass flux by entrainment and detrainment closures adequately represent the vertical anisotropic part of the turbulence. More generally, this study also demonstrates the reliability of EDMF
approaches in representing the ABL. In future work, we plan to evaluate the modified parameterization of the AROME model in

operational mode and with the help of the MAGIC campaign (Kalmus et al., 2015). Additionally, the model could be improved to ensure complete consistency of ABL processes (fluxes and variances) with EDMF. In this way, a prognostic vertical velocity equation may be an improvement, in particular for the energy consistency of the model. Moreover, at subkilometric resolution, the shallow convection scale should be partially resolved by the model, although parameterization up to the hectometric scale is necessary. Currently, the AROME EDMF scheme does not include resolved quantities ($\overline{w}_{\mathrm{resolved}}$, $\overline{\phi'\psi'}_{\mathrm{resolved}}$, etc.) very well, resulting in an inconsistent closure of the parameterization with model core. Efforts should be made to include scale-aware features into the model (Honnert et al., 2020).

*Code and data availability.*  This research uses formatted data from the DEPHY modeling community. The DEPHY-SCM standards and reference case drivers can be found on Github: https://github.com/GdR-DEPHY/DEPHY-SCM. The environment required to run AROME SCM simulations is shared here: https://github.com/romainroehrig/EMS, and the graphical tool: https://github.com/romainroehrig/SCM-atlas. The research model Meso-NH is freely accessible and can be downloaded from its website http://mesonh.aero.obs-mip.fr/mesonh57/. The latest HighTune explorer tool version is available via https://svn.lmd.jussieu.fr/HighTune/trunk/ repository.

*Author contributions.*  A. Marcel contributed to all aspects of this paper, from its conception to editing the manuscript. S. Riette and D. Ricard contributed to the conception and analysis of the study, as well as commenting on the manuscript. C. Lac provided comments on the paper. This work was carried out as part of A. Marcel's PhD, supervised by S. Riette and D. Ricard.

*Acknowledgements.*  We would like to thank Fleur Couvreux and Quentin Rodier for their Meso-NH expertise, as well as the whole DEPHY and HighTune community for their experience and the framework developed around this research.





**Appendix A:  Description of the HighTune Explorer steps**

1. **Build the $i^{\text{th}}$ wave experimental design**: The first step is to define the global design of the wave $i$. For the first wave, we choose a finite number of parameters ($\simeq 10$) related to the AROME physics, not too many because HTexplo is also computationally limited. We provide a range of parameter values, given the uncertainty of parameters values in the literature and the acceptable range for AROME algorithms. It gives a parameter space to explore with the tool. Subsequent waves reuse the refocused parameter space from the previous waves.

2. **Metrics selection**: One of the main counterparts of the tool is that we have to select metrics that should be representative for each ideal case we want to evaluate. This step is difficult and particularly critical because it imposes a subjective choice on which metrics are representative for each case. The metric can be a simple scalar representing either a raw meteorological variable or a more complicated space-time average or profile.

3. **Run experimental design SCM**: This step consists of running a number of SCM simulations with a set of sampled parameters from the current wave parameter space. The number of samples is manually provided to HTexplo. The tool uses Latin hypercubes to sample the space as homogeneously as possible. In fact, the sample size should be as large as possible, but in practice, simulations are performed on the order of magnitude of 10 times the number of selected parameters (Williamson et al., 2017).

4. **Compute metrics SCM vs. LES**: The chosen metrics from both the performed SCM simulations and LES from the Meso-NH model, are calculated at this step.

5. **Building emulator**: This stage aims to predict the values of the selected metrics on the emulator sampled space (sampling different from that used for SCM simulations) with a training set that is the result of step 3. HTexplo uses surrogate models for each computed metric based on emulators. These emulators are statistical models using Gaussian processes, which has the advantage to predict both the values and the uncertainties of the emulated metrics (Salter and Williamson, 2016; Audouin et al., 2021). More detailed information on emulators structures, hypothesis and mathematical framework can be found in reference articles (Couvreux et al., 2021; Williamson et al., 2013).

6. **History matching**: The emulated metrics values are compared with the reference LES. In order to mitigate the problem of LES uncertainties mentioned at step 2, a set of LES has been run preliminarily for each case, taking into account external variabilities (lateral forcing, physics, etc.). It provides a reference LES variance $\sigma_{r,f}^2$ for a metric $f$. HTexplo is based on the history matching technique, which searches for "plausible" and "implausible" values of parameters from the statistical information provided by the emulator. If $\lambda$ is a set of model parameters, then if we know, for an emulated metric value $f(\lambda)$, the surrogate model behavior, i.e., the expectation $\mathrm{E}[f(\lambda)]$ and the uncertainty $\mathrm{Var}[f(\lambda)]$, the reference (LES) error $\sigma_{r,f}^2$ and the SCM discrepancy $\sigma_{d,f}^2$ (the intrinsic inability of the model to reproduce the LES),





for the reference LES value of the metric $r_f$, then the implausibility definition is:

$$I_f(\lambda) = \frac{|r_f - \mathrm{E}[f(\lambda)]|}{\sqrt{\sigma_{r,f}^2 + \sigma_{d,f}^2 + \mathrm{Var}[f(\lambda)]}}$$

The implausibility $I_f(\lambda)$ is the distance between the surrogate model expectation value $\mathrm{E}[f(\lambda)]$ to the LES value of the metric $f$. At the end of the wave $i$, the Not Ruled Out Yet space $\mathrm{NROY}_f^i$ for a metric is then defined as:

$$\mathrm{NROY}_f^i = \{\lambda_i | I_f(\lambda_i) < T_i\}$$

where $T_i$ is a tolerance chosen to exclude any parameter values for which the emulated values are too far away from the LES. The intrinsic uncertainty of LES is a limit to the accuracy of the processes we want to reproduce in the model through parameterizations. Thus, minimizing the distance between the metrics of a 1D model and those of an LES can lead to over-tuning problems (Williamson et al., 2017). As a result, it is better to reject metrics that are too far from a reference than to minimize the distance to a reference.

7. **Parameter space refocusing**: For multiple metrics and cases, the remaining space is the intersection of all spaces for each metric $f_k$. An additional tolerance is used for many metrics such that the remaining space is:

$$\mathrm{NROY}^i = \{\lambda_i | \#\{k | I_{f_k}(\lambda_i) > T_i\} \leq \tau_i\}$$

where $\tau_i$ is a number of metrics for which the model is allowed to be far from the reference and # refer to the number of metrics. Therefore, all these steps reduce and refocus the original parameter space selected at wave $i$ for the wave $i + 1$.



## Appendix B:  TKE budget



**Figure B1.** Different terms of TKE resolved budget $[m^2 s^{-3}.10^5]$ for the FIRE case: thermal production (first line), shear sinks/sources (second line), turbulent transport (third line), dissipation (fourth line), and pressure correlation (fifth line) for the LES (first column), the CTRL experiment (second column), and the NEW AROME version (third column).



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
