# Peer review of "An update of shallow cloud parameterization in the AROME NWP model"

_EGUsphere, 2025_

## Referee Comment (RC2)

Review of "An update of shallow cloud parameterization in the AROME NWP model" by Marcel et al.

**Summary:**

This study updated several moist physical parameterization schemes within the AROME NWP model to improve shallow cloud simulation. The work includes evaluations using both Single Column Model (SCM) simulations and Large Eddy Simulationx (LES) simulations across four distinct cloud cases. It also incorporated a semi-automatic parameter tuning tool to enhance model performance. The updated model shows promising improvements in several key variables, such as cloud fraction, cloud and rain water content, and turbulent kinetic energy.

The manuscript presents a thorough account of the modifications and their impacts. To strengthen the paper's scientific contribution and better align it with the scope of ACP, I recommend restructuring the narrative to more clearly highlight the scientific questions and the novelty of the approach. For instance, the paper could focus on one or two key modifications and deeply explore the underlying physical processes.

**Major comments:**

The manuscript documents a significant number of modifications across several physical schemes. However, it is challenging for the reader to quantitatively assess the specific contribution of each individual modification to the final simulated improvements. The final evaluation of the new AROME configuration includes the cumulative effect of all physical scheme updates plus the parameter tuning from the HTexplo tool. To help the community better understand the physical mechanisms driving the improvements, I strongly suggest a more detailed breakdown.

Specifically, it would be extremely valuable to see a 'Tuning vs. Physics' analysis. This could be achieved by showing a comparison between the control run, a run with all the physical scheme modifications but without the HTexplo tuning, and the final new configuration. This would clearly separate the contributions of the new physics from the new tuning.

Additionally, to further enhance the paper's scientific impact, consider adding a section or a supplementary figure that systematically shows the impact of one or two of the key modifications on the relevant cloud variables. For example, a "Figure 10-13"-style plot that shows the incremental changes from the control run as each major modification is added would be highly informative. This would make the scientific significance of each update much more apparent and provide a clearer path for other researchers looking to adopt similar techniques.

**Other comments:**

- Line 5: "the associated precipitation" could be rephrased as "the cloud microphysical scheme" for greater specificity.
- Line 10: "a transition case" could be more clearly described as "a stratocumulus-to-cumulus case."
- Line 25: It appears "in" is missing before "Wyngaard (2004)"
- Line 30: To provide a broader context, consider including citations for other HOC schemes, such as the CLUBB scheme used in CESM2 and E3SM models
- Line 55: The final sentence in this paragraph appears to shift topics abruptly. To improve the flow, please ensure the discussion of radiation and microphysics is more smoothly integrated or moved to a more suitable section.
- Line 60: between Couvreux et al. (2021) and Hourdin et al. (2021): replace semicolon by comma
- Line 110: It would be helpful to briefly explain what input profiles and large-scale forcings are used and how they are generated.
- Consider adding a table to summarize the four cloud cases, including their cloud type and time period, for easier reference.
- Section 2.2: To clarify the methodology, please explain the difference between the AROME and Meso-NH models as they are used in the study. Additionally, please specify the horizontal and vertical resolutions used for the Meso-NH LES simulations.
- Line 170: "where $B_u$ is strong and detrain…": please change "detrain" to "detrains".
- Line 205: in the formula for $\bar{s'}_{ED}^2$, I believe it should be "$\bar{d}^2 * \bar{T'}^2$. For clarity, it would also be helpful to show the formulas for CF and $\bar{r_c}$.
- Figure 2: The LES line shows discontinuous characteristics. A note explaining the cause of this, such as the conditional sampling method, would be helpful to readers.
- Line 305: "fractionnal" → "fractional"
- Line 325: It would be helpful to define the parameters alpha and beta directly within the text rather than solely referring to previous studies.
- Line 330: Using 'w' for both "updraft" and "wet" can be confusing. Consider using a different variable, like 'wet' or 'cld,' to distinguish them.
- Figure 5: Consider directly plotting the TKE to more clearly show the improvement between Equation 24 and Equation 22.
- Line 455: The definition of Max(CF) is unclear. Please provide a clear definition.
- Figure 8's caption: "HTexplo experience" → "HTexplot experiment"? The same applies to Table 1's caption.

- Figure 9: The "Remaining space" in the bottom right of the figure could be explained in the figure caption to improve clarity.
- L625: While the model shows improvements, the claim that it "can accurately reproduce" cloud fractions and cloud water content might be overstated. Figure 13, for example, highlights several areas where discrepancies remain. To maintain scientific precision, I suggest revising this sentence to acknowledge both the successes and the remaining limitations.

---

## Author Comment (AC1)

We would like to thank RC1 and RC2 for their thorough reading of our manuscript and for the valuable feedback they provided.

We address all comments in detail below; the corrections made are shown in blue.

**General comments**

The authors present a study detailing updates to the AROME turbulence, shallow convection, cloud and microphysics schemes in an effort to improve the representation of shallow clouds and turbulent transport in the model. As tools, they choose large eddy simulations of four canonical shallow cloud cases to provide a reference, and test their model changes in a single column version of AROME. The High Tune Explorer is used to optimize the parameter space of the updated model.

How to adequately model turbulence and shallow convection at kilometer (and increasingly sub-kilometer) scales is an unresolved question that urgently needs answering as operational regional forecasting takes place at these higher resolution, fully entering the turbulent gray zone. I therefore find this study, which aims to address this topic in a framework that treats turbulence and shallow convection in a consistent manner, to be timely and appropriate for publication in ACP.

The model changes described consist of several incremental upgrades to existing schemes, rather than entirely new parameterizations, but this is generally the strategy pursued in operational NWP and does not, in my opinion, detract from the study's relevance.

While I think that all the pieces are there, I would like to see the authors restructure and clarify their discussion (see specific comments below).

**Specific comments**

One weakness of the authors' chosen methodology – comparing individual "golden day" LES and SCM simulations – is that model improvements seen for these idealized cases do not always translate into model improvements in the real world where conditions are rarely "ideal". Including the SANDU transition case is a start. I don't mind the authors' strategy of focusing on a single case to demonstrate the individual parameterization upgrades, but I would like to see more emphasis put on demonstrating that the updates discussed in the context of one case are of benefit to all four cases (or not).

We agree that the SCM versus LES methodology has intrinsic limitations. Firstly, as you mentioned, the SCM cases are simulated using a framework of highly idealized conditions, including the stratocumulus-to-cumulus transition. Secondly, and most importantly, it does not account for interactions with the dynamical core of the model, which also provides the parameterized physics with real-world conditions. However, developing parameterizations is numerically expensive in a real 3D model configuration. We have added a sentence in the introduction that clearly states this limitation.

A few places where this is not obvious are:

1. The authors state as a main motivating factor that AROME has large radiative biases associated with low clouds. Together with cloud fraction, the next most important cloud property for the cloud radiative effect is likely the condensate amount. The authors show improved liquid water content for the RICO case (Fig. 7d) from the new cloud scheme, but

we don't see the impact on LWP/cloud condensate for all four cases after the full implementation and parameter optimization (Sec. 4). While doing a radiation evaluation may be outside the scope of this paper, it would still be good to get a clearer picture how CF and LWP combine to potentially improve the radiation bias for the final version of the model. This would close the circle from motivation to conclusion regarding the radiation bias.

We did not focus specifically on the radiation scheme. The bias in the radiative budget has been observed in 3D. Part of this bias was later attributed to low clouds. The next two figures show the cloud water mixing ratio (mg.kg-1) and LWP (g.m-2) for all cases. We have included the LWP figure in Appendix C and a discussion sentence in Section 4. We decided not to include the cloud water mixing ratio figure in the paper because it shows strongly correlated results with the cloud fraction figure in Section 4. As suggested by the reviewer, we have followed his wise advice to loop back in the conclusion to the radiation bias introduced in the introduction section.

And the figure of LWP (included in the manuscript appendices), the red, black and green curves refer to the LES, AROME NEW and AROME CTRL experiments respectively. For the SANDU case, LES originates from the SAM model (see in the next comment for the explanation).

2. Why is the SANDU case not included in Fig. 13? To me, this case is of special interest as it includes both the well-mixed Sc case, as well as an increasingly decoupled Cu under Sc boundary layer towards the end of the simulation. Many models struggle exactly with this "in between" state, and a good performance here would demonstrate promise that the model changes will also lead to improvements in less idealized settings. The SANDU case is not included in any of the in-depth discussions in section 3, so I would like to see more emphasis put on this case at least in the discussion of the final model configuration after the parameter optimization. The total TKE in Fig. 13 would be great, a decomposition as in Fig B1 would be even better, to see if/how the balance of TKE contribution terms shifts as the BL decouples.

As suggested by the reviewer, we have added the SANDU case to the figure showing the TKE evolution in Section 4. However, due to technical issues, we were unable to (re)perform the LES with the Meso-NH model. LES has been performed on this case earlier with other models (SAM, DALES, UCLA and DHARMA), demonstrating a similar development of the decoupled ABL (see in the following figure). We used the SAM model, as this is the one that is closest to the Meso-NH physics. We have clarified the differences between LES in Section 2.1 and added a discussion sentence in Section 4. Furthermore, we plotted the budget contribution to the TKE for the SANDU case in the second figure, but did not include it in the paper due to constraints on the number of figures.

Temporal evolution of the cloud fraction (first line), cloud water mixing ratio (second line), rain water mixing ratio (third line) and TKE (last line) for the SAM (first column), DALES (second column), DHARMA (third column) and UCLA (last column) LES models. Overall, the main difference between these LES appears to be in the representation of precipitation.

Figure showing TKE sources and sinks (note that the second line color bar scale is 5 times smaller than the others. The layout of the figure is similar to Figure D1).

**Additional discussion points:**

• Sec. 3.2.1 What is the impact of removing the small updraft fraction assumption? The next figure shows an example of small updraft fraction removal for the ARMCu case, averaged between 19:00 and 21:00 UTC. The green, blue and red curves refer to the LES, AROME NEW and CTRL experiments, respectively. The impact is quite limited (although some authors argue that it does have an impact in real model configurations). For this reason, we decided not to include the figure in the manuscript. However, we added a comment about this weak effect in section 3.2.1.

• It appears that the greatest impact on the diurnal cycle of stratocumulus was due to the limitation on the entrainment (not to exceed detrainment), which was removed as a preliminary step (Eq. 7, Sec. 3.1). The improved diurnal cycle is brought up in a few places as a major outcome of the model upgrades (L12 abstract, L620 conclusion), but receives hardly any discussion throughout the paper. If this fix is considered to be a major improvement, then I think the authors should discuss in a bit more detail how the old limiter impacted the EDMF scheme negatively, and what the MF transport looks like without the limiter. So far, there's exactly half a sentence of explanation on L248. Alternatively, if the authors consider this to be on the level of a bug fix or algorithmic fix then this change should be de-emphasized as a main outcome of this study (though the impact is large enough I'd go with the first option).

This particular change only affects entrainment and detrainment within the cloud layer. The first formulation of the shallow convection scheme originates from Pergaud et al. (2009), who applied the buoyancy sorting mechanism of Kain and Fritsch (1990), which was later modified by Bretherton et al. (2004). However, it includes this specific "modification" without any justification. For stratocumulus boundary layers, this causes the updraft to overshoot too far into the free atmosphere, leading to cloud oscillations. An explanatory sentence has been added to section 3.1. Modifications have

**also been made to the abstract and conclusion to emphasize cloud properties rather than the consequences for the diurnal cycle.**

Overall, section 5 (Discussion) does not seem to flow as well as the text in other sections, and needs reworking.

- It starts out with the statement that the original AROME did not match LES energy diagnostics well, and specifically that TKE was too weak for the ED component. The text (L567) suggests that this is a known, previously established problem in AROME. While Sec. 3 shows that the TKE budget is improved by the addition of the anisotropic MF contributions, the systematic TKE underestimate is not previously brought up as a starting point (either in the introduction, or in Sec. 3), so the statement comes as a bit of a surprise in Section 5. I think it would be worth bringing up this point earlier in the paper to motivate the work. √ A sentence mentioning the TKE issue has been added in the section 3.3.1.
- In L 574, it is not entirely clear which bias the statement "It is not clear what causes the TKE bias in Fig. 13" refers to − which experiment? Which specific aspect of the bias? Or do they mean "the remaining bias"? √ The main remaining bias is found in the cloud layer for all four ABL cases. The sentence has been corrected.
- L578: I thought the BL89 length scale was replaced by the adaptive Rodier et al. (2017) length scale (3.2.2)? Why is BL89 mentioned here? Also, the impact of changing the length scale isn't discussed anywhere (and should be). √ It was a typo, we have changed "BL89" to "RM17". We are introducing the RM17 length scale in the manuscript because it is included in the physical pack. However, it is difficult to highlight its impact on the cases. We have performed other 1D cases (referred to as the AYOTTES and IHOP dry ABL cases, not shown in the manuscript), which demonstrate improvements in the ABL properties where the AROME mass flux is not triggered. We have revised section 3.3.2.
- L581 "We did not use local lengths so." appears to be an incomplete sentence. √ The sentence has been corrected.

L626: "The model can accurately reproduce cloud fractions, cloud water content and turbulence according to LES conditional sampling diagnostics." I find this too strong a statement, after just pointing out in the previous section that there are still some rather large and unresolved errors in the TKE, for example, and little is shown on the improvements in water content (see above comment). A more appropriate statement might be that "the improved model more accurately reproduces cloud fractions etc. .... according to LES conditional sampling diagnostics." There is clear improvement, but still some error.  $\sqrt{\text{We have softened the force of the word used in the conclusion}}$  (accurately  $\rightarrow$  better)

**Technical corrections:**

- Abstract, first sentence: "... for *the* parameterization of the Atmospheric Boundary Layer (ABL)", or leave out "the" and use plural (Atmospheric Boundary Layers). √
- Typo L25: "One" should be capitalized  $\sqrt{\phantom{a}}$
- L42: there's a ";" that doesn't belong here before Tan et al.  $\sqrt{\phantom{a}}$
- L47: mis-spelling of "entrainement"  $\sqrt{\phantom{a}}$

- L47: It may be advantageous to use the word "lateral" at least once at the beginning of this discussion of entrainment/detrainment to make clearer what type of entrainment is meant (given that top-entrainment for stratocumulus is another place where entrainment is uncertain). ✓ We agree that the definitions of 'entrainment' and 'detrainment' are confusing. To make it clearer, we have added 'lateral'.
- L59: maybe better computationally "expensive" rather than "intensive"?  $\sqrt{\phantom{a}}$
- L67: "limited-area" sounds better than "area-limited" ✓
- L116: might be good to add here an approximate model layer thickness in the BL √ A sentence has been added.
- Figure 1, and following figures: I find the y-axis (height) on the time-height cross section plots a bit confusing: Sometimes units of metres are used, other times kilometers. For the ARM Cu case, it appears the vertical axis refers to height above mean sea level (or a reference geoid), rather than to height above ground. I would suggest the authors choose consistent units, and show "height above ground" on the y-axis. ✓ The figures y-axis is now showing units of "height above ground [km]". Also, Figure 7 has been corrected.
- Sec. 2.2.5 Liquid water content is referred to as r\_c in this section, later on, it is referred to as q\_l (caption Fig. 7). Please use a consistent naming convention for the variables. √ We have chosen the mixing ratio variables.
- L316: What is meant by a "nearing environment"? √ The « nearing environment » refers to
  the environment surrounding the plume object of the EDMF decomposition. We have
  revised the sentence.
- Eqn. 16: The second term in the MAX function has a minus sign here, but doesn't in Eqn 4. Is that a typo? √ Yes, we have corrected Eqn 4.
- L320: The sentence "The wet part is further complicated to model." sounds incomplete. Do the authors mean "The wet part is more complicated to model."? ✓ **Yes, we have corrected the sentence.**
- L332: You probably mean "upward part", rather than "upper part"? √ Corrected
- L514: The acronym MUSC isn't introduced anywhere. √ Corrected
- The section numbering in section 4 is confusing. Sec. 4.3 contains only a single sentence. Should 4.4 be a sub-section of 4.3? √ **Corrected**

---

## Author Comment (AC2)

We would like to thank RC1 and RC2 for their thorough reading of our manuscript and for the valuable feedback they provided.

We address all comments in detail below; the corrections made are shown in blue.

Review of "An update of shallow cloud parameterization in the AROME NWP model" by Marcel et al.

**Summary:**

This study updated several moist physical parameterization schemes within the AROME NWP model to improve shallow cloud simulation. The work includes evaluations using both Single Column Model (SCM) simulations and Large Eddy Simulationx (LES) simulations across four distinct cloud cases. It also incorporated a semi-automatic parameter tuning tool to enhance model performance. The updated model shows promising improvements in several key variables, such as cloud fraction, cloud and rain water content, and turbulent kinetic energy. The manuscript presents a thorough account of the modifications and their impacts. To strengthen the paper's scientific contribution and better align it with the scope of ACP, I recommend restructuring the narrative to more clearly highlight the scientific questions and the novelty of the approach. For instance, the paper could focus on one or two key modifications and deeply explore the underlying physical processes.

**Major comments:**

The manuscript documents a significant number of modifications across several physical schemes. However, it is challenging for the reader to quantitatively assess the specific contribution of each individual modification to the final simulated improvements. The final evaluation of the new AROME configuration includes the cumulative effect of all physical scheme updates plus the parameter tuning from the HTexplo tool. To help the community better understand the physical mechanisms driving the improvements, I strongly suggest a more detailed breakdown.

Specifically, it would be extremely valuable to see a 'Tuning vs. Physics' analysis. This could be achieved by showing a comparison between the control run, a run with all the physical scheme modifications but without the HTexplo tuning, and the final new configuration. This would clearly separate the contributions of the new physics from the new tuning. Additionally, to further enhance the paper's scientific impact, consider adding a section or a supplementary figure that systematically shows the impact of one or two of the key modifications on the relevant cloud variables. For example, a "Figure 10-13"-style plot that shows the incremental changes from the control run as each major modification is added would be highly informative. This would make the scientific significance of each update much more apparent and provide a clearer path for other researchers looking to adopt similar techniques.

We are aware that the figures presented in Section 3 only provide a general overview of the modifications made for specific cases and model variables. The main problem we face is that showing each change at once does not always result in an improvement in trends (potential temperature, humidity, TKE) and cloud representation (cloud fraction, liquid water content, precipitation) in the model for all 1D ABL cases without a parameter calibration. In addition, some modifications introduce different closures compared to the CTRL version. This makes it more difficult to compare certain parameters before and after the modifications are added. The following figure illustrates the 'AROME NEW' experiment with and without parameter calibration. It shows that the uncalibrated version with the modifications is less satisfactory than the CTRL version (for the cloud fraction here).

Without parameter calibration, the successive addition of modifications would lead to a gradual deterioration in the representation of the ABL cases. Therefore, the ideal solution would be to (re)calibrate the entire physics after each modification is implemented to ensure that the set of parameters is plausible in relation to the LES reference. Furthermore, the version of Htexplo is not sufficiently optimized from a numerical point of view. For example, the ten waves used in this manuscript required several days of calculations on a dozen CPUs. Finally, we are also concerned about the constraints in terms of the number of figures.

**Other comments:**

- Line 5: "the associated precipitation" could be rephrased as "the cloud microphysical scheme" for greater specificity. √ We have corrected the sentence.
- Line 10: "a transition case" could be more clearly described as "a stratocumulus-to-cumulus case." √ **Corrected.**
- Line 25: It appears "in" is missing before "Wyngaard (2004)" √ Corrected.
- Line 30: To provide a broader context, consider including citations for other HOC schemes, such as the CLUBB scheme used in CESM2 and E3SM models. √ As the reviewer suggested, we have modified the sentence to include the general CLUBB parameterization.
- Line 55: The final sentence in this paragraph appears to shift topics abruptly. To improve the flow, please ensure the discussion of radiation and microphysics is more smoothly integrated or moved to a more suitable section. ✓ We have revised this part of the introduction.
- Line 60: between Couvreux et al. (2021) and Hourdin et al. (2021): replace semicolon by comma.
  √ Corrected
- Line 110: It would be helpful to briefly explain what input profiles and large-scale forcings are used and how they are generated.
- Consider adding a table to summarize the four cloud cases, including their cloud type and time period, for easier reference.

- Section 2.2: To clarify the methodology, please explain the difference between the AROME and Meso-NH models as they are used in the study. Additionally, please specify the horizontal and vertical resolutions used for the Meso-NH LES simulations. ✓ In order to respond to the reviewer's advice to clarify the methodology, we have addressed the three previous comments by completing and reworking sections 2.1 and 2.2. In addition, we have added a table in Appendix A, which provides a general description of the ABL cases used, their initial profiles and large-scale forcings, as well as the LES configurations used.
- Line 170: "where B\_u is strong and detrain...": please change "detrain" to "detrains". √
- Line 205: in the formula for \bar{s'\_{ED}^2}, I believe it should be "\bar{d}^2 \* \bar{T'}^2. For clarity, it would also be helpful to show the formulas for CF and \bar{r\_c}. ✓ The square and the formulations of CF and r c from CB02 scheme have been added.
- Figure 2: The LES line shows discontinuous characteristics. A note explaining the cause of this, such as the conditional sampling method, would be helpful to readers. √ We have added a sentence to section 3.2.2 that specifies this behaviour using the conditional sampling method.
- Line 305: "fractionnal" → "fractional" √ Corrected
- Line 325: It would be helpful to define the parameters alpha and beta directly within the text rather than solely referring to previous studies. √ We have added the definitions of Alpha and Beta in the section 3.2.3.
- Line 330: Using 'w' for both "updraft" and "wet" can be confusing. Consider using a different variable, like 'wet' or 'cld,' to distinguish them. ✓ We have replace the subscript 'w' by 'm' for the 'moist' part.
- Figure 5: Consider directly plotting the TKE to more clearly show the improvement between Equation 24 and Equation 22. √ As suggested by the reviewer, the temporal evolution of the TKE (similarly to manuscript figure 5) is illustrated with the following figure. As we previously explained with regard to parameters calibration, adding a modification without re-calibrating the model's physics can lead to a deterioration in prognostic trends for the wrong reasons. Figure 5 of the manuscript clearly shows an improvement in the transport term using equation 23 rather than equation 25, contrary to the figure showing the temporal evolution of TKE below. This deterioration is linked to error compensation, probably due to an excessively high dissipation coefficient. In this case, we therefore prefer to keep figure 5 of the manuscript.

- Line 455: The definition of Max(CF) is unclear. Please provide a clear definition. √ **The definition has been made clearer.**
- Figure 8's caption: "HTexplo experience" → "HTexplot experiment"? The same applies to Table 1's caption. √ We have replaced all the "experience" by "experiment".
- Figure 9: The "Remaining space" in the bottom right of the figure could be explained in the figure caption to improve clarity. √ We have added an additional sentence in the figure caption.
- L625: While the model shows improvements, the claim that it "can accurately reproduce" cloud fractions and cloud water content might be overstated. Figure 13, for example, highlights several areas where discrepancies remain. To maintain scientific precision, I suggest revising this sentence to acknowledge both the successes and the remaining limitations. ✓ We have softened the force of the word used in the conclusion (accurately → better)